# EXPLORING DIFFUSION MODELS' CORRUPTION STAGE IN FEW-SHOT FINE-TUNING AND MITIGATING WITH BAYESIAN NEURAL NETWORKS

## ABSTRACT

Few-shot fine-tuning of Diffusion Models (DMs) is a key advancement, significantly reducing training costs and enabling personalized AI applications. However, we explore the few-shot fine-tuning dynamics of DMs and observe an unanticipated phenomenon: during the fine-tuning process, image fidelity initially improves, then unexpectedly deteriorates with the emergence of noisy patterns, only to recover later with severe overfitting. We term the stage with generated noisy patterns as *corruption stage*. To understand this corruption stage, we begin by heuristically modeling the one-shot fine-tuning scenario, and then extend this modeling to more general cases. Through this modeling, we identify the primary cause of this corruption stage: a narrowed learning distribution inherent in the nature of few-shot fine-tuning. To tackle this, we apply Bayesian Neural Networks (BNNs) on DMs with variational inference to implicitly broaden the learned distribution, and present that the learning target of the BNNs can be naturally regarded as an expectation of the diffusion loss and a further regularization with the pretrained DMs. This approach is highly compatible with current few-shot fine-tuning methods in DMs and does not introduce any extra inference costs. Experimental results demonstrate that our method significantly mitigates corruption, and improves the fidelity, quality and diversity of the generated images in both object-driven and subject-driven generation tasks. The code is available at an anonymous link[1].

## 1 INTRODUCTION

Recent years have witnessed a surge in the development of Diffusion Models (DMs). These models have showcased extraordinary capabilities in various applications, such as image editing (Kawar et al., 2022) and video editing (Yang et al., 2022), among others. Particularly noteworthy is the advent of few-shot fine-tuning methods (Hu et al., 2021; Ruiz et al., 2023; Qiu et al., 2023), in which a pretrained model is fine-tuned to personalize generation based on a small set of training images. These approaches have significantly reduced both memory and time costs in training. Moreover, these techniques offer powerful tools for adaptively generating images based on specific subjects or objects, embodying personalized AI and making AI accessible to everyone. In recent years, this innovation has even fostered the emergence of several communities, such as `civitai.com`, which boasts tens of thousands of checkpoints and millions of downloads.

Despite the importance and widespread usage of few-shot fine-tuning methods in DMs, these methods often struggle or even fail when transferring from a large distribution (i.e., pretrained DMs' learned distribution) to a much smaller one (i.e., fine-tuned DMs' learned distribution) using limited data (Qiu et al., 2023; Ruiz et al., 2023). We are the first to identify when and how these failures occur, and find that they are related to an unusual phenomenon: As shown in Fig. 1, the similarity between the generated images and the training images initially increases during fine-tuning, but then unexpectedly decreases, before increasing once more. Ultimately, the DMs are only capable of generating images identical to the training images. Notably, on the stage with decreasing similarity, we observe there appear some unexpected noisy patterns on the generated images, therefore we name it as *corruption stage*.

---

[1] `https://anonymous.4open.science/r/BNN-Finetuning-DMs-0C35`

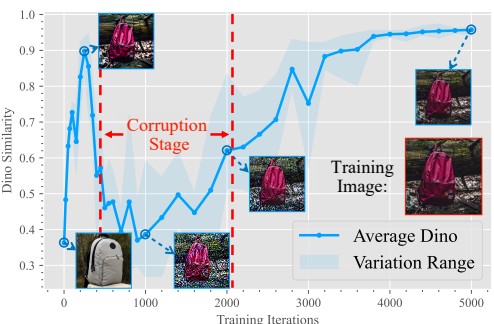

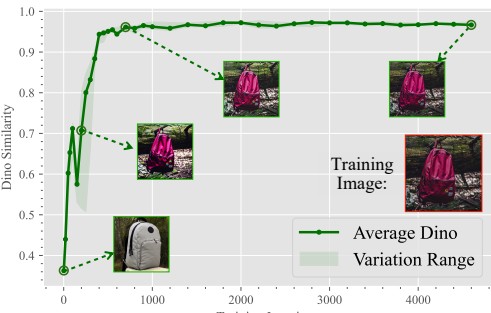

(a) Few-shot fine-tuning process without BNNs.  (b) Few-shot fine-tuning process with BNNs.

Figure 1: Image fidelity variation during few-shot fine-tuning with and without BNNs. Zero training iteration indicates pretrained DMs. We fine-tune Stable Diffusion v1.5 with DreamBooth for 5 runs.

To understand this corruption stage, we carry out further analysis on the few-shot fine-tuning process. Specifically, we start from heuristic modeling a one-shot case, i.e., using only one image during fine-tuning, and then extend it to more general cases. The modeling provides an estimation of the error scale remaining in the generated images. We further show how corruption stages emerge due to the limited learned distribution inherent in few-shot tasks with this modeling.

Based on the analysis above, the solution to corruption should concentrate on expanding the learned distribution. However, this expansion remains challenging in few-shot fine-tuning. Traditional data augmentation methods, when applied to generative models, often face significant problems such as leakage (Karras et al., 2020) and a reduction in generation quality (Daras et al., 2023). Inspired by recent advancements in Bayesian Neural Networks (BNNs) (Blundell et al., 2015), we propose to incorporate BNNs as a straightforward yet potent strategy to implicitly broaden the learned distribution. We further present that its learning target can be regarded as an expectation of the diffusion loss and an extra regularization loss related to the pretrained model. These two losses can be adjusted to reach a trade-off between image fidelity and diversity. Our method does not introduce any extra inference costs, and has good compatibility with existing few-shot fine-tuning methods in DMs, including DreamBooth (Ruiz et al., 2023), LoRA (Hu et al., 2021), and OFT (Qiu et al., 2023). Experimental results demonstrate that our method significantly alleviates the corruption issues and substantially enhances the performance across various few-shot fine-tuning methods on diverse datasets under different metrics.

In summary, our main contributions are as follows:

1. We observe an abnormal phenomenon during few-shot fine-tuning process on DMs: The image fidelity first enhances, then unexpectedly worsens with the appearance of noisy patterns, before improving again but with severe overfitting. We refer to the phase where noisy patterns appear as the corruption stage. We hope this observation could inform future research on DMs.

2. We provide a heuristic modeling for few-shot fine-tuning process on DMs, explaining the emergence and disappearance of the corruption stage. With this modeling, we pinpoint that the main issue stems from the constrained learned distribution of DMs inherent in few-shot fine-tuning process.

3. We innovatively incorporate BNNs to broaden the learned distribution, hence relieving such corruption. Experimental results confirm the effectiveness of this approach in improving on different metrics, including text prompt fidelity, image fidelity, generation diversity, and image quality.

## 2 RELATED WORKS

### 2.1 DIFFUSION MODELS AND FEW-SHOT FINE-TUNING

Diffusion Models (DMs) (Ho et al., 2020; Sohl-Dickstein et al., 2015; Song & Ermon, 2019; Song et al., 2020) are generative models that approximate a data distribution through the gradual denoising of a variable initially sampled from a Gaussian distribution. These models involve a forward diffusion process and a backward denoising process. In the forward process, the extent of the addition of a noise $\varepsilon$ increases over time $t$, as described by the equation $x_t = \sqrt{\alpha_t}x_0 + \sqrt{1 - \alpha_t}\varepsilon$, where $x_0$ is a given original image and the range of time $t$ is $\{1, \ldots, 1000\}$ in general cases. Conversely, in

the backward process, the DMs aim to estimate the noise with a noise-prediction module $\epsilon_\theta$ and subsequently remove it from the noisy image $x_t$. The discrepancy between the actual and predicted noise serves as the training loss, denoted as diffusion loss $\mathcal{L}_{DM} := \mathbb{E}_{\varepsilon \sim \mathcal{N}(0,1), t} \left\| \epsilon_\theta(x_t, t) - \varepsilon \right\|_2^2$.

Few-shot fine-tuning in DMs (Gal et al., 2022; Hu et al., 2021; Qiu et al., 2023; Ruiz et al., 2023) aims at personalizing DMs with a limited set of images, facilitating the creation of tailored content. Gal et al. (2022) introduced a technique that leverages new tokens within the embedding space of a frozen text-to-image model to capture the concepts presented in provided images. Ruiz et al. (2023) further proposed DreamBooth, which fine-tunes most parameters in DMs leveraging a reconstruction loss that captures more precise details of the input images, with a class-specific preservation loss ensuring the alignment with the textual prompts. Additionally, Hu et al. (2021) proposed LoRA, a lightweight fine-tuning approach that inserts low-rank layers to be learned while keeping other parameters frozen. Qiu et al. (2023) presented OFT, a method that employs orthogonal transformations to enhance the quality of generation. Although these methods generally succeed in capturing the details of training images, they suffer from the corruption stage observed in this paper.

## 2.2 BAYESIAN NEURAL NETWORKS

Bayesian Neural Networks (BNNs) are a type of stochastic neural networks characterized by treating the parameters as random variables rather than fixed values (Blundell et al., 2015; Buntine, 1991; Neal, 2012). The objective is to infer the posterior distribution $P(\theta|\mathcal{D})$ for the parameters $\theta$ given a dataset $\mathcal{D}$. It endows BNNs with several distinct advantages, such as the capability to model the distributions for output, to mitigate overfitting, and to enhance model interpretability (Arbel et al., 2023; Jospin et al., 2022). One prevalent variant of BNNs is the mean-field variational BNN, also known as *Bayes by Backprop*, where the mean-field variational inference is applied to obtain the variational distribution $Q_W(\theta)$ to approximate the posterior distribution $P(\theta|\mathcal{D})$ (Blundell et al., 2015). Recent studies demonstrated that even a BNN module, which treats only a subset of parameters as random variables while maintaining the rest as fixed, can retain the benefits associated with full BNNs (Harrison et al., 2023; Kristiadi et al., 2020; Sharma et al., 2023). Our proposed method can be regarded as a natural progression of BNN principles applied to few-shot fine-tuning in DMs.

## 3 CORRUPTION STAGE IN FEW-SHOT FINE-TUNING

In Sec. 3.1, we first present the observation on the corruption stage during few-shot fine-tuning of DMs. In Sec. 3.2, to better observe and understand the issues and fine-tuning dynamics with the corruption stage, we propose a heuristic modeling that uses Gaussian distribution as an approximation. To address the challenges of modeling the dynamics of DMs during fine-tuning, we adopt reasonable simplifications, supported by evidence from a specific case. In Sec. 3.3, we explain the emergence and vanishing of the corruption stage by our modeling, and reveal the limited learned distribution is the root cause of the corruption stage.

### 3.1 OBSERVATION

In this section, we explore the performance variation during the few-shot fine-tuning process of DMs. Concretely, we fine-tune Stable Diffusion (SD) v1.5[2] (Rombach et al., 2022) with DreamBooth (Ruiz et al., 2023) on different number of training images, and record the average Dino similarity between the generated images and training images as a measure of the image fidelity (Qiu et al., 2023; Ruiz et al., 2023).

As shown in Fig. 2, the variation of the image fidelity is not monotonous during fine-tuning. Specifically, the few-shot fine-tuning process can be approximately divided into the following phases:

1. In the first a few iterations, the image fidelity improves quickly.
2. Later, there is an **abnormal decrease** in the image fidelity. We observe the generated images in this phase present with increasing noisy patterns, i.e., the gradual emergence of the corruption stage.

---

[2]https://huggingface.co/runwayml/stable-diffusion-v1-5

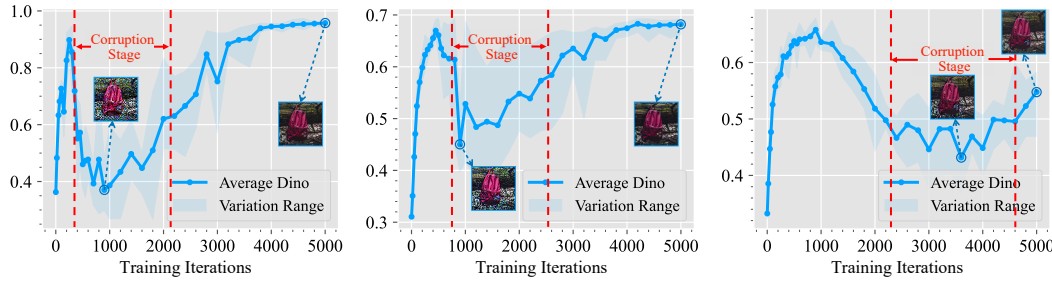

(a) Fine-tuning on 1 image.    (b) Fine-tuning on 2 images.    (c) Fine-tuning on 6 images.

Figure 2: Illustration of image fidelity variation in few-shot fine-tuning under different numbers of training images measured by Dino similarity. Higher Dino similarity indicates better image fidelity. As the number of training images increases, the corruption occurs later, and its severity is reduced.

3. Subsequently, the generation fidelity recovers, and we observe the corruption patterns on generated images progressively diminish, i.e. the gradual disappearance of the corruption stage. Once corruption completely diminishes in this phase, the model enters a state close to *overfitting*, where it can only generate images identical to the training images. As a result, it loses the ability to produce diverse images.

Through the comparisons between Fig. 2a, Fig. 2b and Fig. 2c, it is worth noting that when the number of training images increases, the onset of corruption is delayed and its severity is lessened.

### 3.2 HEURISTIC MODELING ON FEW-SHOT FINE-TUNING IN DMS

In this section, we begin by heuristically modeling one-shot fine-tuning scenarios and then extend it to more general cases. This modeling is supported by an evidence on a specific case.

**Heuristic Modeling for One-shot Fine-tuning on DMs.** We start with a representative condition where the dataset $\mathcal{D}$ contains only one training image $x'$. Under this condition, we suppose the fine-tuned DMs with parameter $\theta$ model the joint distribution of any original image $x_0$ and any noisy image at time $t$, i.e., $x_t$, as a multivariate Gaussian distribution $P_\theta(x_0, x_t)$. Concretely, its marginal distribution of $x_0$ is approximated as $P_\theta(x_0) \approx \mathcal{N}(x', \sigma_1^2)$ when the model is fine-tuned with only one image $x'$. Additionally, as the noisy image $x_t$ is obtained by a linear combination $x_t = \sqrt{\alpha_t}x_0 + \sqrt{1 - \alpha_t}\epsilon$ between $x_0$ and a unit Gaussian noise $\epsilon$, the marginal distribution of $x_t$ should approximate $P_\theta(x_t) \approx \mathcal{N}(\sqrt{\alpha_t}x', \alpha_t\sigma_1^2 + (1 - \alpha_t))$. Notably, the fine-tuning process in fact narrows the KL divergence between $P_\theta(x_t \mid x_0 = x')$ and $\mathcal{N}(\sqrt{\alpha_t}x', (1 - \alpha_t))$, thus these distributions should be increasingly close during fine-tuning (Qiu et al., 2023; Ruiz et al., 2023).

With this modeling, the main focus of the DMs, predicting the original image $x_0$ based on a noisy image $x_t$, is represented as $P_\theta(x_0|x_t)$. We find that it in fact approximates a Gaussian distribution related to both $x_t$ and the training image $x'$:

$$P_\theta(x_0|x_t) \approx \mathcal{N}(x' + \delta_t(x_t, x'), \frac{(1 - \alpha_t)}{\alpha_t\sigma_1^2 + (1 - \alpha_t)}),$$

$$\text{where } \delta_t(x_t, x') = \frac{\sqrt{\alpha_t}\sigma_1^2}{\alpha_t\sigma_1^2 + (1 - \alpha_t)}(x_t - \sqrt{\alpha_t}x'). \tag{1}$$

The most possible $x_0$ denoted as $\hat{x}_0$ in the view of the DMs is:

$$\hat{x}_0 = \arg\max_{x_0} P_\theta(x_0|x_t) \approx x' + \delta_t(x_t, x'). \tag{2}$$

The derivation details are provided at the Appendix Sec. A. Notably, the error term $\delta_t(x_t, x')$ represents the difference between the predicted original image $\hat{x}_0$ and the training image $x'$. Intuitively, $\sigma_1$ can be treated as the "confidence" of the fine-tuned DM regenerating the training sample $x'$.

The accuracy of the above modeling is influenced by the number of training iterations. As fine-tuning progresses, the approximation $P_\theta(x_0) \approx \mathcal{N}(x', \sigma_1^2)$ becomes more precise, and the formulation more closely reflects real-world scenarios. To illustrate an extreme case in this modeling, we consider a scenario where $\sigma_1 = 0$, i.e., $\delta_t = 0$. Under this condition, for any input $x_t$, the DMs consistently

reproduce the training image $x'$ as described by Eq. (2). This indicates that the DMs fully lose the intrinsic denoising ability in this extreme scenario, only regenerating the training image instead.

In the opposite extreme, where $\sigma_1 = +\infty$, i.e., $\delta_t = \frac{1}{\sqrt{\alpha_t}} x_t - x'$, the model's prediction for $x_0$ is exactly $\frac{1}{\sqrt{\alpha_t}} x_t$. This indicates that the DMs entirely lose the ability to generate images. Instead, DMs only rescale $x_t$ based on the factor $\alpha_t$ at the time step $t$, leading to any noise in $x_t$ is also remaining in the generated image.

**Extension to More General Cases.** We further extend our modeling to the case where $\mathcal{D}$ contains multiple training samples. In specific, we assume the learned distribution of the original image $x_0$ of the DMs, i.e., $P_\theta(x_0)$, is centered with an image set $\mathcal{I}_\theta$. Under few-shot fine-tuning, as the training continues, $\mathcal{I}_\theta$ gradually approximates the training dataset $\mathcal{D}$. On the other hand, for the pretrained DMs, they typically learn a much larger manifold, which can be interpreted as learning with a sufficiently large $\mathcal{I}_\theta$. For all these DMs facing with noisy image $x_t$, we simplify their behavior as firstly finding a sample $x^* \in \mathcal{I}_\theta$ to minimize the error term $\delta_t(x_t, x^*)$, and then estimating the corresponding $\hat{x}_0$ according to Eq. (2). With this simplification, the sufficiently large $\mathcal{I}_\theta$ of pretrained DMs enables them to find an $x$ such that the error term $\delta_t(x_t, x)$ approaches zero, thereby preventing corruption in most instances.

**Support for the Modeling.** To support the above modeling closely approximates the practical scenarios, we present a specific example where we set the "noisy" image $x_t = 0$, and then make both the pretrained and fine-tuned DMs denoise this image $x_t$ which is entirely free of noise. This is a special case as $x_t$ is free of noise and the typical DM should leave it unchanged to function effectively as a denoiser. According to our modeling, both DMs should first find one sample $x^*$ within their own $\mathcal{I}_\theta$. Naturally, $x_t = 0$ is within $\mathcal{I}_\theta$ of the pretrained DMs (See Appendix Sec. E for more evidence). Therefore, the pretrained DMs should leave this $x_t = 0$ almost unchanged during denoising.

In comparison, the $\mathcal{I}_\theta$ of the fine-tuned DMs should gradually approximate the training dataset $\mathcal{D}$. Therefore, the fine-tuned DMs should first find one sample $x^* \in \mathcal{D}$, and then predict the original image $x_0$ as proportional to $x^*$ according to Eq. (2).

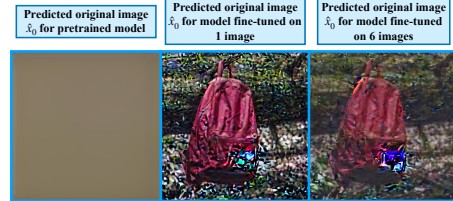

Figure 3: Denoised images from pretrained and fine-tuned DMs using $x_t = 0$ and $t = 1000$. The pretrained DMs do not largely change $x_t$ as it is free of noise. In contrast, both DMs fine-tuned on 1 and 5 images transform $x_t$ to make it closely resembles one sample within the training dataset $\mathcal{D}$.

Experimental results provided in Fig. 3 support our analysis, where we can observe the pretrained DMs do not largely change this $x_t = 0$, but the fine-tuned DMs transform this $x_t$ to one of the samples in its training dataset $\mathcal{D}$. This supports that our modeling aligns with real-world scenarios during few-shot fine-tuning.

### 3.3 Explanation of the Corruption Stage.

In this section, we explain the corruption stage according to our heuristic modeling about few-shot fine-tuning process on DMs. We first present the reason for the emergence of the corruption stage, with an example showing the severity of such problem. We further demonstrate why the corruption stage gradually disappears, resulting in "overfitting" as the fine-tuning process continues.

**Emergence of the Corruption Stage.** As stated in Eq. (2), given any noisy image $x_t$, the fine-tuned DMs would predict the original image $\hat{x}_0 = x^* + \delta_t$, where $x^* \in \mathcal{I}_\theta \approx \mathcal{D}$ after certain training iterations. The scale of the error term $\delta_t$ is related to $\left\| x_t - \sqrt{\alpha_t} x^* \right\|_2$ and $\sigma_1$. We estimate $\sigma_1$ based on $x_t$ sampled from $\mathcal{N}(0, 1 - \alpha_t)$ and present the average $\sigma_{1,t}$ among different $t$s in Fig. 4. It shows that the $\sigma_1$ remains relatively high under moderate iterations, resulting in a significant $\delta_t$ once $x_t$ is not identical to $\sqrt{\alpha_t} x^*$.

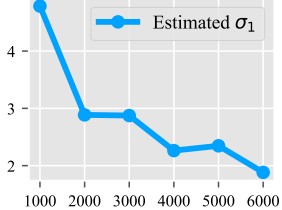

Figure 4: Estimated $\sigma_1$ under different training iterations for DreamBooth.

Concretely, for the case with only one training image, i.e., $\mathcal{D} = \{x'\}$, we set the noisy image $x_t = \sqrt{\alpha_t}x' + \sqrt{1-\alpha_t}\varepsilon + \delta'$, where $\varepsilon \in \mathcal{N}(0,1)$ and $\delta'$ is an additional noise introduced to a small region of the image. This $\delta'$ simulates the case where the generating process of DMs is inaccurate in some $t$s. We further set the time variable $t = 100$, and fine-tune DMs with 1000 iterations, with the estimated $\sigma_1 \approx 4.8$ as shown in Fig. 4. According to Eq. (2) and the analysis above, we can compute $\|\delta_{100}\|_2^2 \approx 2.65 \|\delta'\|_2^2$. It means the additional noise $\delta'$ introduced is even

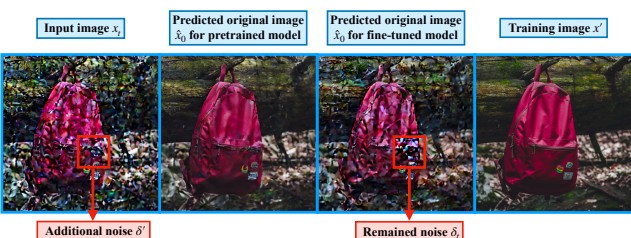

Figure 5: Experimental results for pretrained and fine-tuned DMs with an additional noise $\delta'$ introduced in a small region of the noisy image $x_t$ at $t = 100$. The pretrained DMs effectively remove $\delta'$, producing high-quality images. Conversely, the fine-tuned DMs fail to eliminate $\delta'$, with output images showing corruption patterns.

expanded in this case, leading to a significant error term $\delta_t$. Fig. 5 shows the experimental results under this setting, where we observe a significant error term $\delta_t$, resembling a corruption pattern, and consisting with the above analysis.

**Vanishing of the Corruption Stage.** However, with the fine-tuning continues, $\sigma_1$ drops as shown in Fig. 4, leading to a decreasing prediction error $\delta_t$. This indicates that the corruption stage vanishes, and the fine-tuned DMs gradually move to the state where they only strictly regenerate the training image $x' \in \mathcal{D}$. This is a classic case of "overfitting", where the fine-tuned DMs lose their ability to generate diverse outputs and thus become unusable.

In conclusion, the analysis in this Sec. 3.3 shows how the corruption stage happens when the learned distribution of DMs is highly limited with small $\mathcal{I}_\theta$ and a high standard deviation $\sigma_1$.

## 4 APPLYING BNNs TO FEW-SHOT FINE-TUNING ON DMs

### 4.1 MOTIVATION

Based on our analysis, the corruption stage primarily arises from a limited learned distribution with a small $\mathcal{I}_\theta$. Motivated by recent work on BNNs (Blundell et al., 2015; Harrison et al., 2023; Kristiadi et al., 2020; Sharma et al., 2023), which model the parameters $\theta$ as random variables, we propose to apply BNNs in the few-shot fine-tuning process on DMs as a simple yet effective method to expand $\mathcal{I}_\theta$. Intuitively, the modeling of BNNs hinders the DMs to learn the exact distribution of the training dataset $\mathcal{D}$. Therefore, the DMs are encouraged to learn a larger and more robust distribution to counter the randomness. Moreover, the sampling randomness during the fine-tuning process can be regarded as an inherent data augmentation, indicating the corresponding $\mathcal{I}_\theta$ is implicitly expanded.

### 4.2 FORMULATION

**Modeling.** BNNs model the parameters $\theta$ as random variables. Therefore, the learned distribution of DMs with BNN is $P(x|\mathcal{D}) = \int P(x|\theta)P(\theta|\mathcal{D})d\theta$. Concretely, $P(x|\theta)$ is the image distribution modeled by the DMs, and $P(\theta|\mathcal{D})$ is the posterior parameter distribution with a given dataset $\mathcal{D}$.

As the posterior distribution $P(\theta|\mathcal{D})$ is intractable, a variational distribution $Q_W(\theta)$ is applied to approximate it. We model the variational distribution of each parameter $\theta$ as a Gaussian distribution: $\theta \sim \mathcal{N}(\mu_\theta, \sigma_\theta^2)$, where $W = \{\mu_\theta, \sigma_\theta\}$ are trainable parameters. Considering the fine-tuning process, we initialize the expectation term $\mu_\theta$ from the corresponding parameter of the pretrained DMs, denoted as $\theta_0$. Following previous work (Blundell et al., 2015), we apply the re-parameterization trick to obtain the gradients of the parameters, as detailed in Appendix Sec. C.

**Training.** During the fine-tuning process, the DMs are trained by minimizing the Kullback–Leibler (KL) divergence $\mathrm{KL}(Q_W(\theta)||P(\theta|\mathcal{D}))$, which is equivalent to minimizing

$$\mathcal{L} = -\int Q_W(\theta) \log \frac{P(\theta, \mathcal{D})}{Q_W(\theta)} d\theta = \mathbb{E}_{\theta \sim Q_W(\theta)} \underbrace{-\log P(\mathcal{D}|\theta)}_{\mathcal{L}_{DM}} + \underbrace{\mathrm{KL}(Q_W(\theta)||P(\theta))}_{\mathcal{L}_r}. \quad (3)$$

Following previous work (Zhang et al., 2022), the above loss $\mathcal{L}$ can be divided into two terms. In DMs, the first term can be seen as the modeled probability for the training dataset $\mathcal{D}$, and is equivalent to the expectation of the diffusion loss $\mathcal{L}_{DM}$ shown in Sec. 2.1 on the parameters $\theta$. The second term can be seen as a regularization restricting the discrepancy between the variational distribution $Q_W(\theta)$ and the prior distribution $P(\theta)$. We name it as the regularization loss $\mathcal{L}_r$. In few-shot fine-tuning, we regard the pretrained DMs naturally represent the prior information, so we set the prior distribution $P(\theta)$ from the pretrained DMs, i.e., $P(\theta) = \mathcal{N}(\theta_0, \sigma^2)$, where $\sigma$ is a hyperparameter which represents the parameter randomness.

In practice, we formulate our learning target as a linear combination of $\mathcal{L}_{DM}$ and $\mathcal{L}_r$ with a hyperparameter $\lambda$, i.e.,

$$W^* = \arg\min_W \mathbb{E}_{\theta \sim Q_W(\theta)} \mathcal{L}_{DM} + \lambda \mathcal{L}_r. \quad (4)$$

The training process is summarized in Appendix Alg. 1. Empirically, we find that using only $\mathbb{E}_{\theta \sim Q_W(\theta)} \mathcal{L}_{DM}$, i.e., setting $\lambda$ as 0, is enough to improve few-shot fine-tuning. Nevertheless, we can reach a further trade-off between the generation diversity and image fidelity by adjusting $\lambda$.

**Inference.** During the inference, we explicitly replace each parameter $\theta$ with its mean value $\mu_\theta$ and perform inference just as DMs without BNNs. It guarantees that we do not introduce any additional costs compared to fine-tuned DMs without BNNs when deployed in production.

Motivated by previous approaches on BNN modules (Harrison et al., 2023; Kristiadi et al., 2020; Sharma et al., 2023), we only model a subset of parameters as random variables in practice, which reduces the computational costs. Fine-tuning DMs with BNNs is compatible with existing few-shot fine-tuning methods, including DreamBooth (Ruiz et al., 2023), LoRA (Hu et al., 2021), and OFT (Qiu et al., 2023). More details are presented in Appendix Sec. D.

## 5 EXPERIMENTS

We apply BNNs to different few-shot fine-tuning methods across different tasks. For object-driven generation, where the fine-tuned DMs synthesize images with the details of given objects, we use all the 30 classes from DreamBooth (Ruiz et al., 2023) dataset, each containing 4-6 images. For subject-driven generation, where the fine-tuned DMs synthesize images with the identities of given people, we follow previous research (Van Le et al., 2023), randomly selecting 30 classes of images from CelebA-HQ (Liu et al., 2018), each containing 5 images. Most training settings follow previous approaches (Hu et al., 2021; Qiu et al., 2023; Ruiz et al., 2023; Van Le et al., 2023). All experiments are conducted with 5 different seeds by default and we report the average performance here. We use Stable Diffusion v1.5[3] (SD v1.5) as the default model for fine-tuning. As for the BNNs, we set the default initialized standard derivation $\sigma_\theta$ and prior standard derivation $\sigma$ as 0.01. The $\lambda$ is set as 0 by default. We show more details in Appendix Sec. H.1.

Following previous approaches (Hu et al., 2021; Qiu et al., 2023; Ruiz et al., 2023), for each class, we fine-tune one DM and generate 100 images with various prompts. These generated images are used to measure the performance of different few-shot fine-tuning methods. In specific, we use Clip-T (Radford et al., 2021) to measure the text prompt fidelity, Clip-I (Hessel et al., 2021) and Dino (Caron et al., 2021) to assess image fidelity, and Lpips (Zhang et al., 2018) to evaluate generation diversity. Additionally, we apply Clip-IQA (Wang et al., 2023) to measure no-reference image quality. We show more details about these metrics in Appendix Sec. G.

### 5.1 QUANTITATIVE AND VISUALIZED COMPARISONS

We apply BNNs on different few-shot fine-tuning methods under both object-driven and subject-driven generation tasks. As shown in Tab. 1 and Fig. 6, BNNs bring considerable improvements

---

[3]https://huggingface.co/runwayml/stable-diffusion-v1-5

Table 1: Performance of fine-tuning with BNNs under object-driven and subject-driven generation.

| Object-Driven Generation: DreamBooth Dataset | | | | | | Subject-Driven Generation: CelebA Dataset | | | | |
|---|---|---|---|---|---|---|---|---|---|---|
| Method | Clip-T↑ | Dino↑ | Clip-I↑ | Lpips↑ | Clip-IQA↑ | Method | Clip-T↑ | Dino↑ | Clip-I↑ | Lpips↑ | Clip-IQA↑ |
| DreamBooth | 0.246 | 0.614 | 0.771 | 0.611 | 0.875 | DreamBooth | 0.186 | 0.642 | 0.723 | 0.511 | 0.789 |
| DreamBooth w/ BNNs | **0.256** | **0.633** | **0.785** | **0.640** | **0.893** | DreamBooth w/ BNNs | **0.205** | **0.696** | **0.757** | **0.515** | **0.811** |
| LoRA | 0.252 | 0.542 | 0.722 | 0.650 | 0.864 | LoRA | 0.216 | 0.602 | 0.656 | 0.644 | 0.804 |
| LoRA w/ BNNs | **0.261** | **0.618** | **0.769** | **0.678** | **0.890** | LoRA w/ BNNs | **0.227** | **0.604** | **0.663** | **0.688** | **0.824** |
| OFT | 0.233 | 0.649 | 0.786 | 0.629 | 0.861 | OFT | 0.164 | 0.675 | 0.728 | 0.549 | 0.784 |
| OFT w/ BNNs | **0.242** | **0.661** | **0.791** | **0.646** | **0.884** | OFT w/ BNNs | **0.185** | **0.696** | **0.743** | **0.570** | **0.798** |

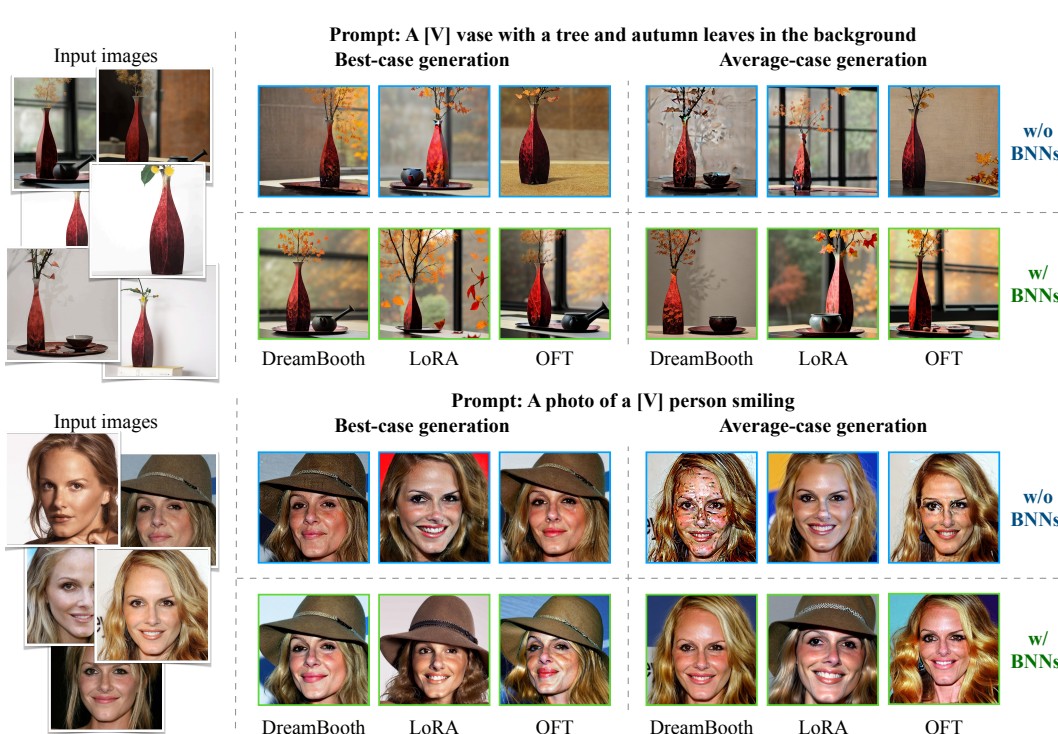

Figure 6: Comparison of few-shot fine-tuning methods with and without BNNs across subject-driven and object-driven scenarios. We show both best-case and average-case generated images measured by Clip-I, Dino and Clip-IQA. See Appendix Sec. I and L for more details.

on all few-shot fine-tuning methods across text prompt fidelity (Clip-T) and image fidelity (Dino and Clip-I). These improvements arise from the expanded learned distribution contributed by BNNs, which makes the DMs more capable to generate reasonable images about the learned objects/subjects based on different prompts. BNNs also largely enhance the no-reference image quality (Clip-IQA). It is mainly because BNNs largely reduce the corruption phenomenon, which also partially improve the image fidelity (Dino and Clip-I) as the corrupted images are semantically distant to training images. Additionally, we observe BNNs boost the generation diversity (Lpips). We believe it naturally comes from the randomness introduced by BNNs.

## 5.2 GENERALIZATION

In this section, we further demonstrate BNNs can be applied to broader scenarios with notable performance improvement, including different DMs, varying training steps and a different number of training images. By default, we experiment on DreamBooth with BNNs for fine-tuning.

Table 2: Performance under different DMs. All DMs are fine-tuned with DreamBooth on DreamBooth dataset.

| | | Clip-T↑ | Dino↑ | Clip-I↑ | Lpips↑ | Clip-IQA↑ |
|---|---|---|---|---|---|---|
| SD v1.5 | w/o BNNs | 0.246 | 0.614 | 0.771 | 0.611 | 0.875 |
| | w/ BNNs | **0.256** | **0.633** | **0.785** | **0.640** | **0.893** |
| SD v1.4 | w/o BNNs | 0.248 | 0.594 | 0.762 | 0.618 | 0.872 |
| | w/ BNNs | **0.249** | **0.620** | **0.777** | **0.656** | **0.895** |
| SD v2.0 | w/o BNNs | 0.240 | 0.563 | 0.739 | 0.604 | 0.875 |
| | w/ BNNs | **0.248** | **0.610** | **0.764** | **0.649** | **0.925** |

**Different DMs.** Following previous work (Ye et al., 2024), we experiment on different DMs. Concretely, besides default SD v1.5, we also experiment on SD v1.4 and SD v2.0 (Rombach et al.,

2022). We provide training details in Appendix Sec. H.2. Tab. 2 indicates that applying BNNs on different DMs of SD consistently improves few-shot fine-tuning across multiple metrics.

**Training Iterations.** Fig. 7 shows that our method consistently improves the image quality (Clip-IQA). It also improves the image fidelity (Dino) when the training steps are larger than $100 \times$ Num, where Num represents the number of images utilized during fine-tuning. With fewer iterations, the model suffers from underfitting. In this case, BNNs may make the underfitting problem further severe as BNNs encourage the model to learn a larger distribution. This results in the slightly decreased image fidelity (Dino) observed in $100 \times$ Num.

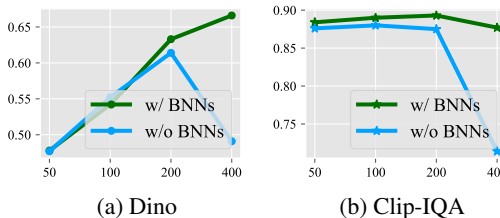

(a) Dino      (b) Clip-IQA

Figure 7: Comparison of performance with and without BNNs on the DreamBooth dataset with different training iterations per image.

**Numbers of Training Images**. We also experiment under different numbers of training images. Concretely, we use the CelebA-HQ (Liu et al., 2018) dataset as it contains enough images per class. We randomly select five classes from the CelebA-HQ dataset and conduct experiments with different numbers of training images. We fix the training iterations to $250 \times$ Num.

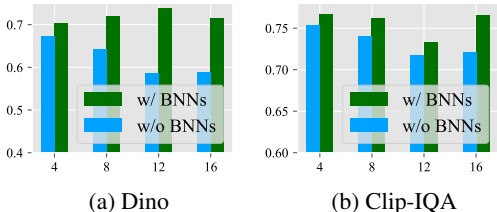

(a) Dino      (b) Clip-IQA

Figure 8: Comparison of the performance with different number of training images.

Fig. 8 illustrates that our method consistently improves image fidelity and quality under different numbers of training images. Such improvement is more obvious with more training images. This is primarily because the $250 \times$ Num generally results in more severe corruption problems when the training image number Num is larger, hence BNNs bring in much improvement by expanding the learning distribution and relieving corruptions.

## 5.3 ABLATION STUDY

**Scale of Initialized** $\sigma_\theta$**.** The initialized standard deviation $\sigma_\theta$ determines the extent of randomness during fine-tuning. We experiment with applying BNNs under varying initialized $\sigma_\theta$. Experimental results in Fig. 9a indicate that both the image fidelity (Dino) and quality (Clip-IQA) have been improved with a moderate initialized $\sigma_\theta$. However, when the initialized $\sigma_\theta$ is too large, the DMs collapse and the performance rapidly decreases. This indicates the DMs are almost randomly updated because of the too large randomness introduced in this scenario.

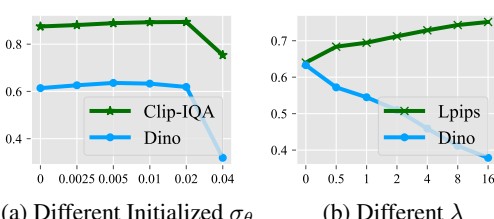

(a) Different Initialized $\sigma_\theta$      (b) Different $\lambda$

Figure 9: Ablation study on different initialized $\sigma_\theta$ and different $\lambda$.

**Trade-off Between Diversity and Fidelity with Adjusted** $\lambda$**.** Eq. (4) indicates $\lambda$ controls the trade-off between the generation diversity and image fidelity. As illustrated in Fig. 9b, an increasing $\lambda$ leads to improving generation diversity (Lpips), albeit at the expense of image fidelity (Dino).

**Where to Apply BNNs.** As mentioned in Sec. 4.2, we could model only a subset of parameters as random variables, i.e., applying BNNs on a part of layers in DMs. By default, we apply BNNs to all linear layers in the U-Net (Ronneberger et al., 2015) except the ones in the cross-attention modules and explore how different choices influence the performance and the training costs.

As shown in Tab. 3, the DM can achieve relatively good performance with only the upblock of the U-Net applied with BNNs, which reduces the ratio of the parameters modified to about $13.8\%$. We can further decrease the training costs by only applying BNNs on the normalization layer, i.e., Layer Normalization (LN) (Ba et al., 2016) and Group Normalization (GN) (Wu & He, 2018) layers. This decreases the ratio of the parameters modified to about $0.02\%$ with relatively strong performance.

Table 3: Comparison of performance when BNNs are applied to different layers in DMs. All experiments are conducted on DreamBooth dataset fine-tuning with DreamBooth. 'N.A.' refers to no BNN applied. 'CrossAttn' refers to BNN applied to cross-attention modules. We report the GPU memory costs and average time costs during fine-tuning for each class on one A100 GPU.

| Layer Type | CrossAttn | Up Block Only | Clip-T↑ | Dino↑ | Clip-I↑ | Lpips↑ | Clip-IQA↑ | Memory Costs(MB) | Time Costs(s) |
|---|---|---|---|---|---|---|---|---|---|
| N.A. | | | 0.246 | 0.614 | 0.771 | 0.611 | 0.875 | 27236 | 933 |
| Linear | | | 0.256 | 0.633 | 0.785 | 0.640 | 0.893 | 31912 | 1107 |
| Linear | | ✓ | 0.259 | 0.614 | 0.776 | 0.646 | 0.890 | 29674 | 1040 |
| Linear | ✓ | | **0.272** | 0.611 | 0.760 | **0.672** | **0.897** | 33258 | 1157 |
| LN+GN | | | 0.254 | **0.650** | **0.792** | 0.643 | 0.889 | 27268 | 1088 |

Table 4: User study results of fine-tuned DMs with and without BNNs across various measurements under both best-case and average-case scenarios. The table depicts the percentage of users favoring generated images from fine-tuned DMs with and without BNNs.

| | Best-case Generation | | | | Average-case Generation | | |
|---|---|---|---|---|---|---|---|
| Method | Subject Fidelity | Text Alignment | Image Quality | Method | Subject Fidelity | Text alignment | Image Quality |
| DreamBooth | 34.4% | 32.3% | 30.2% | DreamBooth | **50.5%** | 35.6% | 28.7% |
| DreamBooth w/ BNNs | **65.6%** | **67.7%** | **69.8%** | DreamBooth w/ BNNs | 49.5% | **64.4%** | **71.3%** |
| LoRA | 48.0% | 34.7% | 31.6% | LoRA | 27.5% | 26.5% | 24.5% |
| LoRA w/ BNNs | **52.0%** | **65.3%** | **68.4%** | LoRA w/ BNNs | **72.5%** | **73.5%** | **75.5%** |
| OFT | 30.6% | 34.7% | 40.8% | OFT | 41.1% | 26.8% | 39.3% |
| OFT w/ BNNs | **69.4%** | **65.3%** | 59.2% | OFT w/ BNNs | **58.9%** | **73.2%** | **60.7%** |

Additionally, when BNNs are applied to cross-attention modules, there is a significant increase in text prompt fidelity (Clip-T) at the expenses of image fidelity (Dino and Clip-I). Intuitively, this is because the input image aligns with only a limited set of prompts, and the applying of BNNs in cross-attention modules produces a further broader distribution matching more prompts.

## 5.4 USER STUDY

**User Study Setting.** We conduct user study to comprehensively present the superiority of applying BNNs on few-shot fine-tuning DMs. Concretely, we follow previous approaches (Qiu et al., 2023; Ruiz et al., 2023) and conduct a structured human evaluation for generated images, involving 101 participants. For this purpose, we utilize the DreamBooth dataset and the CelebA-HQ dataset. Subsequently, we generate four images per subject or object using a random prompt selected from 25 prompts across five models trained with different seeds. For comprehensive comparisons, we select both the best and average cases from the generated images (see Appendix Sec. I for details).

Each participant is requested to compare image pairs generated by models that have been fine-tuned with and without BNNs, using three baseline models: DreamBooth, LoRA, and OFT. For each task, there are three binary-selection questions:

- **Subject fidelity:** Which of the given two images contains a subject or object that is most similar to the following reference image (one from the training dataset)?
- **Text alignment:** Which of the given two images best matches the text description provided below (the prompt used to generate the images)?
- **Image quality:** Which of the given two images exhibits the higher image quality?

**Results.** The results are presented in Tab. 4, which shows the percentage of participants favoring each method (with and without BNNs) based on the criteria described above. It is evident that the methods with BNNs are preferred in almost all scenarios for both best-case and average-case generations. This preference is particularly significant in terms of text alignment and overall image quality.

## 6 CONCLUSION

In this paper, we focus on few-shot fine-tuning in DMs, and reveal an unusual "corruption stage" where image fidelity first improves, then deteriorates due to noisy patterns, before recovering. With theoretical modeling, we attribute this phenomenon to the constrained learned distribution inherent in few-shot fine-tuning. By applying BNNs to broaden the learned distribution, we mitigate the corruption. Experiment results across various fine-tuning methods and datasets underscore the versatility of our approach.

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

## A    DERIVATION OF EQ. (2)

We first restate our assumptions formally. For convenience, we use equal signs instead of approximate signs in our assumptions and derivations.

- The joint distribution of $x_0$ and $x_t$ is modeled by the DM as a multivariate Gaussian distribution

$$P_\theta([x_0, x_t]^T) = \mathcal{N}(\boldsymbol{\mu}, \boldsymbol{\Sigma}) = \mathcal{N}([x', \sqrt{\alpha_t}x']^T, \begin{bmatrix} \sigma_1^2 & c \\ c & \alpha_t\sigma_1^2 + (1 - \alpha_t) \end{bmatrix}), \tag{5}$$

  where $c$ represents the unknown covariance between $x_0$ and $x_t$.
- The conditional probability of $x_t$ given $x_0 = x'$ is

$$P_\theta(x_t \mid x_0 = x') = \mathcal{N}(\sqrt{\alpha_t}x', (1 - \alpha_t)). \tag{6}$$

Denote the inverse matrix of $\boldsymbol{\Sigma}$ as

$$\boldsymbol{\Sigma}^{-1} = \begin{bmatrix} \lambda_{11} & \lambda_{12} \\ \lambda_{21} & \lambda_{22} \end{bmatrix} = \frac{1}{|\boldsymbol{\Sigma}|} \begin{bmatrix} \alpha_t\sigma_1^2 + 1 - \alpha_t & -c \\ -c & \sigma_1^2 \end{bmatrix}, \tag{7}$$

where $|\boldsymbol{\Sigma}| = \sigma_1^2(\alpha_t\sigma_1^2 + (1 - \alpha_t)) - c^2$ is the determinant of $\boldsymbol{\Sigma}$. According to the property of a joint Gaussian distribution, the conditional distribution $P(x_t|x_0)$ can be represented as

$$P(x_t|x_0) = \mathcal{N}(\frac{\lambda_{22}\sqrt{\alpha_t}x' + \lambda_{12}x_0 - \lambda_{12}x'}{\lambda_{22}}, \frac{1}{\lambda_{22}}). \tag{8}$$

According to Eq. (6),

$$\frac{1}{\lambda_{22}} = (1 - \alpha_t), \tag{9}$$

which means $c = \pm\sqrt{\alpha_t}\sigma_1^2$. Hence we know the joint distribution is

$$P_\theta([x_0, x_t]^T) = \mathcal{N}(\boldsymbol{\mu}, \boldsymbol{\Sigma}) = \mathcal{N}([x', \sqrt{\alpha_t}x']^T, \begin{bmatrix} \sigma_1^2 & \pm\sqrt{\alpha_t}\sigma_1^2 \\ \pm\sqrt{\alpha_t}\sigma_1^2 & \alpha_t\sigma_1^2 + (1 - \alpha_t) \end{bmatrix}). \tag{10}$$

Repeatedly, according to the property of a joint Gaussian distribution, we have

$$P(x_0|x_t) = \mathcal{N}(x' \pm \frac{\sqrt{\alpha_t}\sigma_1^2(y - \sqrt{\alpha_t}x')}{\alpha_t\sigma_1^2 + (1 - \alpha_t)}, \frac{1 - \alpha_t}{\alpha_t\sigma_1^2 + 1 - \alpha_t}). \tag{11}$$

In practice, the positive sign is more reasonable as it indicates the deviations of the predicted image $x_0$ and the input noisy image $x_t$ are aligned. Therefore,

$$P(x_0|x_t) = \mathcal{N}(x' + \frac{\sqrt{\alpha_t}\sigma_1^2(y - \sqrt{\alpha_t}x')}{\alpha_t\sigma_1^2 + (1 - \alpha_t)}, \frac{1 - \alpha_t}{\alpha_t\sigma_1^2 + 1 - \alpha_t}). \tag{12}$$

## B    PROOF FOR THE ERROR TERM APPROACHES ZERO DURING FINE-TUNING

In this section, we provide a direct proof for the error term approaching zero when the diffusion loss approaches zero.

*Proof.* As previous work points out (Ho et al., 2020), the probability of a diffusion model on a given distribution is lower bounded by its ELBO:

$$-\log P_\theta(x_0) \geq \mathbf{E}_q(x_{1:T}|x_0) \log \frac{P_\theta(x_{0:T})}{Q(x_{1:T}|x_0)},$$

and is simplified to diffusion loss: $L_{DM}(x_0) := \mathbf{E}_{t,\epsilon \in N(0,1)} \|\varepsilon_\theta(\sqrt{\alpha_t}x_0 + \sqrt{1 - \alpha_t}\epsilon, t) - \epsilon\|^2$.

The main target loss of fine-tuning methods is to directly minimize the diffusion loss for data within the fine-tuning data distribution. This can be interpreted as minimizing the KL divergence $\mathbf{E}_Q(x'_{1:T}|x'_0) \log \frac{P_\theta(x'_{0:T})}{Q(x'_{1:T}|x'_0)}$, where $Q(x'_0)$ represents the fine-tuning data distribution.

Under optimal conditions where $L_{DM}(x'_0) \to 0$, we have $P_\theta(x'_0) \to 1$ and $P_\theta(x'_{t+1}|x'_0) \to Q(x'_{t+1}|x'_0)$. This implies the identity of the learned $P$ and fine-tuned data distribution $Q$, hence we have

$$\arg\max P_\theta(x_0|x_t) = \arg\max Q(x_0|x_t) = x'.$$

It corresponds to $\sigma_1 = 0$ in Eq. (1), leading to $\delta_t = 0$. $\square$

It verifies the correctness of our modeling from another side, hence further supports our theoretical analysis.

## C    DETAILS OF APPLYING BNNs

The training process of applying BNNs in fine-tuning is summarized in Alg. 1. To obtain the gradients of the variational parameters, i.e., gradients of $W = \{\mu_\theta, \sigma_\theta\}$, we apply the commonly used reparameterization trick in BNNs. Concretely, we first sample a unit Gaussian variable $\varepsilon_\theta$ for each $\theta$, and then perform $\theta = \mu_\theta + \sigma_\theta \times \varepsilon_\theta$ to obtain a posterior sample of $\theta$. Hence, the gradients can be calculated by

$$\frac{\partial}{\partial W}\mathcal{L} = \frac{\partial}{\partial W}\left[\mathbb{E}_{Q_W(\theta)}\mathcal{L}_{DM} + \mathcal{L}_r\right] \tag{13}$$

$$= \mathbb{E}_{\varepsilon_\theta \sim \mathcal{N}(0,I)}\left[\frac{\partial\mathcal{L}_{DM}}{\partial\theta}\frac{\partial\theta}{\partial W} + \frac{\partial\mathcal{L}_r}{\partial W}\right]. \tag{14}$$

We refer to the Proposition 1 in previous work (Blundell et al., 2015) for the detailed derivation.

---

**Algorithm 1:** Fine-tuning DMs with BNNs

---

**Input:** Initialized variational parameters $W = \{\mu_\theta, \sigma_\theta\}$, prior distributions $P(\theta) = \mathcal{N}(\theta_0, \sigma^2)$, fine-tuning dataset $\mathcal{D}$, number of fine-tuning iterations $N$, hyperparameter $\lambda$
**Output:** Fine-tuned variational parameters $W = \{\mu_\theta, \sigma_\theta\}$
**for** $i = 0$ **to** $N - 1$ **do**
  Sample $\varepsilon_\theta \sim \mathcal{N}(0, I)$.
  Compute $\theta = \mu_\theta + \varepsilon_\theta \circ \sigma_\theta$.
  Sample $x \in \mathcal{D}$, $t \sim \mathcal{U}(1, 1000)$, noise $\varepsilon_t \sim \mathcal{N}(0, 1)$
  Compute $\mathcal{L}_{DM} = \|\varepsilon_t - \epsilon_\theta(x_t, t)\|^2$.
  Compute $\mathcal{L}_r = \mathrm{KL}(P(\theta)\|\mathcal{N}(\mu_\theta, \sigma_\theta^2))$.
  Compute $\mathcal{L} = \mathcal{L}_{DM} + \lambda\mathcal{L}_r$.
  Backward $\mathcal{L}$ and update $\mu_\theta, \sigma_\theta$.
**end for**

---

## D    APPLYING BNNs ON DIFFERENT FEW-SHOT FINE-TUNING METHODS

We provide code for implementation in anonymous repository available at link `https://anonymous.4open.science/r/BNN-Finetuning-DMs-0C35`.

**Applying BNNs on DreamBooth.**    DreamBooth is a full-parameter fine-tuning method, and is one of the mainstream fine-tuning methods (Ruiz et al., 2023). Therefore, all parameters in DreamBooth can be modeled as the BNNs parameters.

**Applying BNNs on LoRA.**    LoRA (Hu et al., 2021) is a classic lightweight yet effective method for few-shot fine-tuning. In LoRA layers, the weight matrix $\mathbf{W} \in \mathbb{R}^{d \times k}$ is modeled as a sum of fixed weight from the pretrained model and a trainable low-rank decomposition: $\mathbf{W} = \mathbf{W}_0 + \mathbf{BA}$,

Table 5: Prompts used for evaluation. [V] indicates the special token and [object] indicates the type of the object.

| Prompts for object-driven generation | Prompts for subject-driven generation |
| --- | --- |
| a [V] [object] in the jungle | a photo of a [V] person wearing sunglasses |
| a [V] [object] in the snow | a photo of a [V] person with snowflakes in their hair |
| a [V] [object] on the beach | a photo of a [V] person with beachy hair waves |
| a [V] [object] on a cobblestone street | a photo of a [V] person wearing a beret |
| a [V] [object] on top of pink fabric | a photo of a [V] person with a neutral expression |
| a [V] [object] on top of a wooden floor | a photo of a [V] person with a contemplative look |
| a [V] [object] with a city in the background | a photo of a [V] person laughing heartily |
| a [V] [object] with a mountain in the background | a photo of a [V] person with an amused smile |
| a [V] [object] with a blue house in the background | a photo of a [V] person with forest green eyeshadow |
| a [V] [object] on top of a purple rug in a forest | a photo of a [V] person wearing a red hat |
| a [V] [object] wearing a red hat | a photo of a [V] person with a slight grin |
| a [V] [object] wearing a santa hat | a photo of a [V] person with a thoughtful gaze |
| a [V] [object] wearing a rainbow scarf | a photo of a [V] person wearing a black top hat |
| a [V] [object] wearing a black top hat and a monocle | a photo of a [V] person in a chef hat |
| a [V] [object] in a chef outfit | a photo of a [V] person in a firefighter helmet |
| a [V] [object] in a firefighter outfit | a photo of a [V] person in a police cap |
| a [V] [object] in a police outfit | a photo of a [V] person wearing pink glasses |
| a [V] [object] wearing pink glasses | a photo of a [V] person wearing a yellow headband |
| a [V] [object] wearing a yellow shirt | a photo of a [V] person in a purple wizard hat |
| a [V] [object] in a purple wizard outfit | a photo of a [V] person smiling |
| a red [V] [object] | a photo of a [V] person frowning |
| a purple [V] [object] | a photo of a [V] person looking surprised |
| a shiny [V] [object] | a photo of a [V] person winking |
| a wet [V] [object] | a photo of a [V] person yawning |
| a cube shaped [V] [object] | a photo of a [V] person laughing |

where $\mathbf{W}_0 \in \mathbb{R}^{d \times k}, \mathbf{B} \in \mathbb{R}^{d \times r}, \mathbf{A} \in \mathbb{R}^{r \times k}$ with rank $r$ (Hu et al., 2021). In our implementation, we only convert the up matrix $\mathbf{A}$ into random variables, and the down matrix $\mathbf{B}$ is still kept as usual trainable parameters to make $P(\mathbf{W})$ a Gaussian distribution. This design also reduces additional computational costs during training while keeps its effectiveness.

**Applying BNNs on OFT.** OFT is a few-shot fine-tuning method where the weights are tuned only by orthogonal transformations (Qiu et al., 2023). In OFT layers, the weight matrix $\mathbf{W} \in \mathbb{R}^{d \times k}$ is modeled as $\mathbf{W} = \mathbf{R}\mathbf{W}_0$, where $\mathbf{R}$ is guaranteed to be an orthogonal matrix by $\mathbf{R} = (\boldsymbol{I} + 0.5(\mathbf{Q} - \mathbf{Q}^T))(\boldsymbol{I} - 0.5(\mathbf{Q} - \mathbf{Q}^T))^{-1}$, and $\mathbf{Q}$ is the trainable parameter in the original OFT method. To guarantee the orthogonality is not destroyed by random sampling in BNNs, we only convert the trainable parameter $\mathbf{Q}$ into random variables. Therefore, after the above transformation, $\mathbf{R}$ is kept as an orthogonal matrix, and $\mathbf{W}$ is kept as an orthogonalization transformation of the original pretrained $\mathbf{W}_0$.

## E    DETAILS OF THE VALIDATION EXPERIMENTS

**Proof for $x_t = 0$ is within the $\mathcal{I}_\theta$ of the pretrained DMs.** We provide two proofs that $x_t = 0$ is within the $\mathcal{I}_\theta$ of the pretrained DMs. The SD v1.5 is used as the pretrained DM.

- As shown in Fig. 10a, we use the prompt "a simple, solid gray image with no textures or variations" to generate images, and we observe the pretrained DM is capable of generating such images free of noise.

- We use the Img2Img Pipeline[4] provided by diffusers with no prompt provided, i.e., unconditional generation. Then we set the input image as a blank one and the img2img strength as 0.1, which means input noisy image $x_{100} = \alpha_{100}\varepsilon$, where $\varepsilon \in \mathcal{N}(0, 1)$. As shown in Fig. 10b. we can also observe the denoised result is completely free of noise.

---

[4]https://github.com/huggingface/diffusers/blob/main/src/diffusers/pipelines/stable_diffusion/pipeline_stable_diffusion_img2img.py

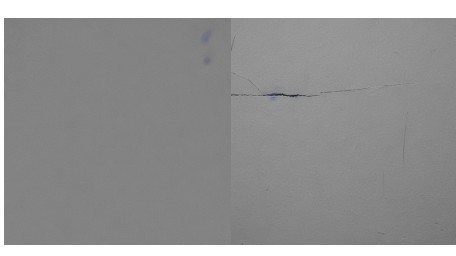
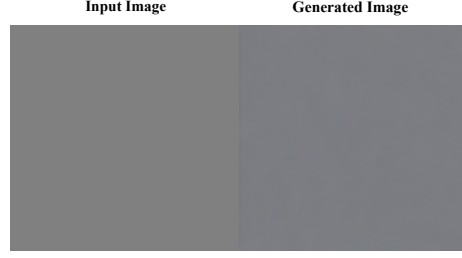

Input Image          Generated Image

(a) Generated Images based on a given prompt.          (b) Img2img result for pure-color image as input.

Figure 10: Proof for $x_t = 0$ is within the $\mathcal{I}_\theta$ of the pretrained DM.

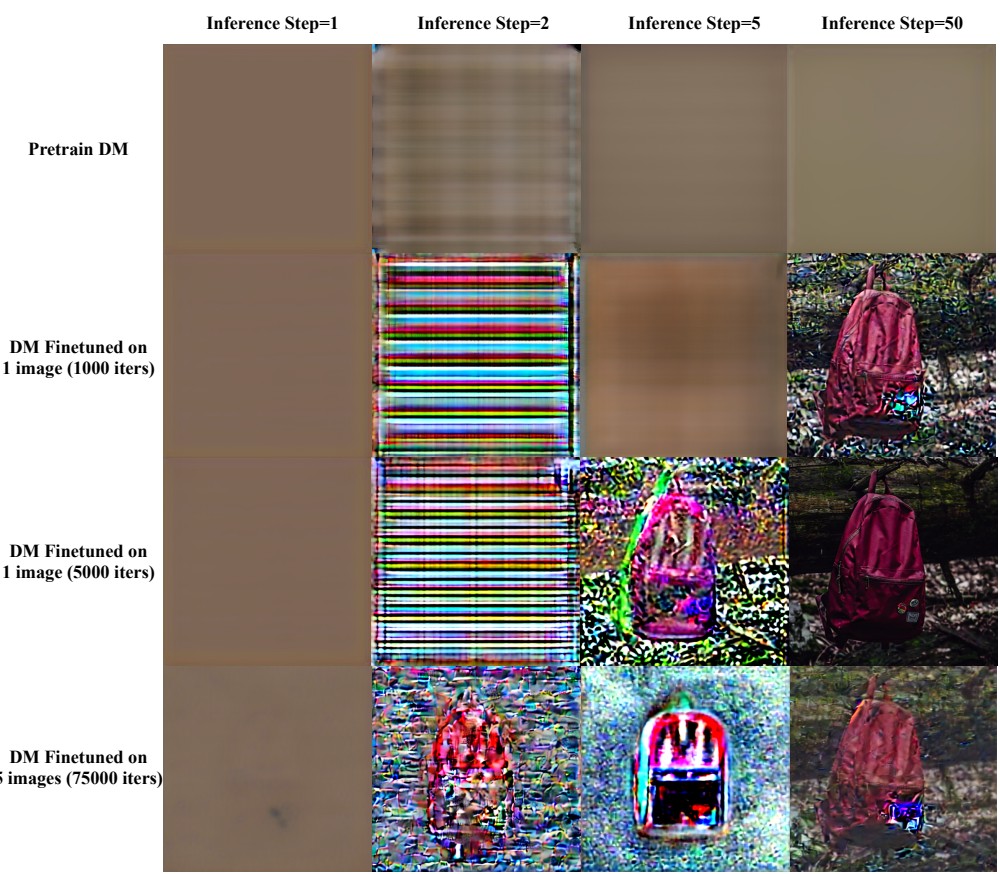

Figure 11: More results for generated images when input $x_t = 0$ under different DMs and different inference steps.

Both results support our argument that $x_t = 0$ is naturally within $\mathcal{I}_\theta$ of the pretrained DMs.

**Settings for Experiments in Sec. 3.3.** We fine-tune an SD v1.5 with DreamBooth without PPL loss (Ruiz et al., 2023). The learning rate is fixed to $5 \times 10^{-6}$ and only the U-Net (Ronneberger et al., 2015) is fine-tuned. We use the backpack class in DreamBooth dataset as an example, and use prompt "a [V] backpack" to train where the "[V]" is the special token. We set $x_{100} = 0$ and show the one-step denoised result using the Img2Img Pipeline provided by diffusers. During denoising, for both pretrained and fine-tuned DMs, the prompt is fixed to "a [V] backpack". More results are shown in Fig. 11.

Table 6: Standard deviation of the results shown in Tab. 1.

| Object-Driven Generation: DreamBooth Dataset | | | | | | Subject-Driven Generation: CelebA Dataset | | | | | |
| --- | --- | --- | --- | --- | --- | --- | --- | --- | --- | --- | --- |
| Method | Clip-T | Dino | Clip-I | Lpips | Clip-IQA | Method | Clip-T | Dino | Clip-I | Lpips | Clip-IQA |
| DreamBooth | 0.0017 | 0.0057 | 0.0037 | 0.0051 | 0.0028 | DreamBooth | 0.0065 | 0.0179 | 0.0115 | 0.0087 | 0.0078 |
| DreamBooth w/ BNNs | 0.0016 | 0.0056 | 0.0035 | 0.0054 | 0.0023 | DreamBooth w/ BNNs | 0.0036 | 0.0102 | 0.0090 | 0.0191 | 0.0095 |
| LoRA | 0.0018 | 0.0087 | 0.0045 | 0.0048 | 0.0084 | LoRA | 0.0034 | 0.0085 | 0.0059 | 0.0060 | 0.0068 |
| LoRA w/ BNNs | 0.0032 | 0.0104 | 0.0064 | 0.0072 | 0.0021 | LoRA w/ BNNs | 0.0025 | 0.0049 | 0.0163 | 0.0078 | 0.0065 |
| OFT | 0.0030 | 0.0063 | 0.0041 | 0.0081 | 0.0045 | OFT | 0.0026 | 0.0194 | 0.0141 | 0.0130 | 0.0115 |
| OFT w/ BNNs | 0.0010 | 0.0028 | 0.0026 | 0.0062 | 0.0068 | OFT w/ BNNs | 0.0024 | 0.0078 | 0.0066 | 0.0061 | 0.0094 |

## F  ADDITIONAL SUPPORT FOR OUR MODELING

To further support our modeling in Sec. 3.2, we present more results with $x_t = kx'$ for different $k$s. When giving the training prompt, according to our modeling in Eq. (2), a fine-tuned DM should predict the original image $\hat{x}_0 = (1 + \frac{\sqrt{\alpha_t}\sigma_1^2}{\alpha_t\sigma_1^2+(1-\alpha_t)}(k - \sqrt{\alpha_t}))x'$, which is a scaling of $x'$.

In comparison, the generated result for the pretrained model should not exhibit a similar correlation with the given training sample $kx'$, as $kx'$ is not part of the pretrained dataset or its training distribution $\mathcal{I}_\theta$. Experimental results shown in Fig. 12 support our analysis, hence further confirming the rationality of our modeling.

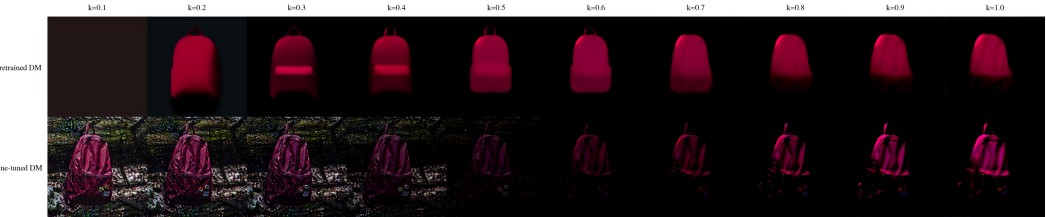

Figure 12: Denoised images from pretrained and fine-tuned DMs using $x_t = kx'$ for different $k$s.

## G  METRICS

We use the following metrics to robustly measure different aspects of the fine-tuned DMs:

**Text prompt fidelity:** Following previous papers (Qiu et al., 2023; Ruiz et al., 2023) , we use the average similarity between the clip (Radford et al., 2021) embeddings of the text prompt and generated images, denoted as Clip-T.

**Image fidelity:** Following previous papers (Qiu et al., 2023; Ruiz et al., 2023), we compute the average similarity between the clip (Radford et al., 2021) and dino (Caron et al., 2021) embeddings of the generated images and training images, denoted as Clip-I (Hessel et al., 2021) and Dino.

**Generation Diversity:** We compute the average Lpips (Zhang et al., 2018) distance between the generated images of the fine-tuned DMs, following previous papers (Qiu et al., 2023; Ruiz et al., 2023).

**Image quality:** We find that corruption largely decreases the visual quality of the images, making them unusable. However, full-reference images quality measurement cannot fully represent this kind of degradation in image quality. Therefore, we propose to add a no-visual quality metric for measurements. We use Clip-IQA (Wang et al., 2023) for measurements, which is one of the SOTA no-reference image quality measurements.

## H  SETTINGS OF FINE-TUNING

We experiment on applying BNNs on different fine-tuning methods with one A100 GPU. All experiments are conducted under all 30 classes with 5 different seeds by default and we report the average performance. The standard deviation of our main result in Tab. 1 is shown in Tab. 6.

## H.1 FEW-SHOT FINE-TUNING HYPER-PARAMETERS

The details of the parameters in the few-shot fine-tuning methods on our default model, i.e., SD v1.5, are presented below. We use $\mathrm{Num}$ to represent the number of images utilized for training.

**Dreambooth**: We use the training script provided by Diffusers[5]. Only the U-Net is fine-tuned during the training process. By default, the number of training steps is set to $200 \times \mathrm{Num}$ on DreamBooth dataset and $250 \times \mathrm{Num}$ on CelebA, with a learning rate of $5 \times 10^{-6}$. The batch size is set to 1, and the number of class images used for computing the prior loss is $200 \times \mathrm{Num}$ by default. The prior loss weight remains fixed at 1.0. For the DreamBooth dataset, the training instance prompt is "a photo of a [V] {class prompt}", where{class prompt} refers to the type of the image (such as dog, cat and so on). For the CelebA dataset, the training instance prompt is "a photo of a [V] person".

**LoRA**: We use the training script provided by Diffusers[6]. All default parameters remain consistent with the case in Dreambooth (No Prior), with the exception of the learning rate and training steps, which are adjusted to $1 \times 10^{-4}$ and to $400 \times \mathrm{Num}$, respectively.

**OFT**: We use the training script provided by the authors[7]. All default parameters remain consistent with the case in Dreambooth (No Prior), with the exception of the learning rate, which is adjusted to $1 \times 10^{-4}$.

For all experiments involving the applying of BNNs, we maintain the default settings for the number of learning steps, learning rate, training prompts, and other hyper-parameters.

For evaluation purposes, each training checkpoint generates four images per prompt, resulting in a total of 100 images from 25 different prompts. The prompts used for evaluation are displayed in Tab. 5, encompassing a broad set to thoroughly assess the variety and quality of the DMs.

## H.2 TRAINING SETTING FOR DMs WITH DIFFERENT ARCHITECTURES

In this section, we provide training settings for applying BNNs on different DMs under DreamBooth. The learning rate is fixed to $5 \times 10^{-6}$ in all cases. The numbers of training iterations are set as $200 \times \mathrm{Num}$ and $400 \times \mathrm{Num}$ for SD v1.4 and SD v2.0, respectively. For BNNs applying on SD v1.4, we set the hyperparameter $\lambda = 0.1$. All other hyperparameters are set as default.

## I BEST-CASE V.S. AVERAGE-CASE GENERATION

In practical scenarios, users typically generate multiple images and manually choose the most suitable one for use. Therefore, comparing the best cases essentially evaluates the quality of the chosen images, while the comparison of average cases assesses the difficulty of selecting an adequate image. Conversely, worst-case scenarios may hold less practical relevance, as users may regenerate images until achieving a satisfactory result, effectively bypassing such cases.

Therefore, we place emphasis on presenting best-case and average-case generation in this paper. Specifically, we first filter the generated images by selecting the top 90% using CLIP-T and Dino to ensure alignment with the prompt and the learned concept. Subsequently, we employ CLIP-IQA to identify both the top-quality and average-quality images. This approach provides a more comprehensive evaluation of the models' performance.

Nonetheless, we also include some comparisons of the worst-case generation results before and after applying BNNs illustrated in Fig. 13 for comprehensiveness. Concretely, we present the images with the lowest image qualities evaluated using CLIP-IQA. Our experiments span 10 random classes in the DreamBooth Dataset, with DreamBooth as the baseline fine-tuning method. The results show that DMs fine-tuned without BNNs usually present the corruption while BNNs successfully mitigate it, leading to better generation quality.

---

[5]https://github.com/huggingface/diffusers/blob/main/examples/dreambooth/train_dreambooth.py
[6]https://github.com/huggingface/diffusers/blob/main/examples/dreambooth/train_dreambooth_lora.py
[7]https://github.com/Zeju1997/oft

Figure 13: Visualization of worst-case generations measured by CLIP-IQA.

## J Limitations and Future Work

Even though few-shot fine-tuning DMs with BNNs applied has shown promising improvements, this paper also introduces a few interesting open problems. Firstly, the extra randomness may make fine-tuning slower. This may lower the generation quality when the DMs are under-fitting. In addition, the ability of learning extremely detailed patterns in the image may be reduced when the number of fine-tuning iterations is insufficient. Future work could focus on these problems.

## K Broader Impact

This paper focuses on advancing few-shot fine-tuning techniques in DMs to provide more effective tools for creating personalized images in various contexts. Previous few-shot fine-tuning methods have faced corruption phenomenon, as mentioned in this paper. Our approach, utilizing BNNs, addresses this phenomenon and provides generated images with higher quality.

However, there is potential for misuse, as malicious entities could exploit these technologies to deceive or misinform. Such challenges underscore the critical need for continuous exploration in this field. The development and ethical application of personalized generative models are not only paramount but also ripe for future research.

## L More Visualizations

We show more visualized results in Fig. 14 and Fig. 15.

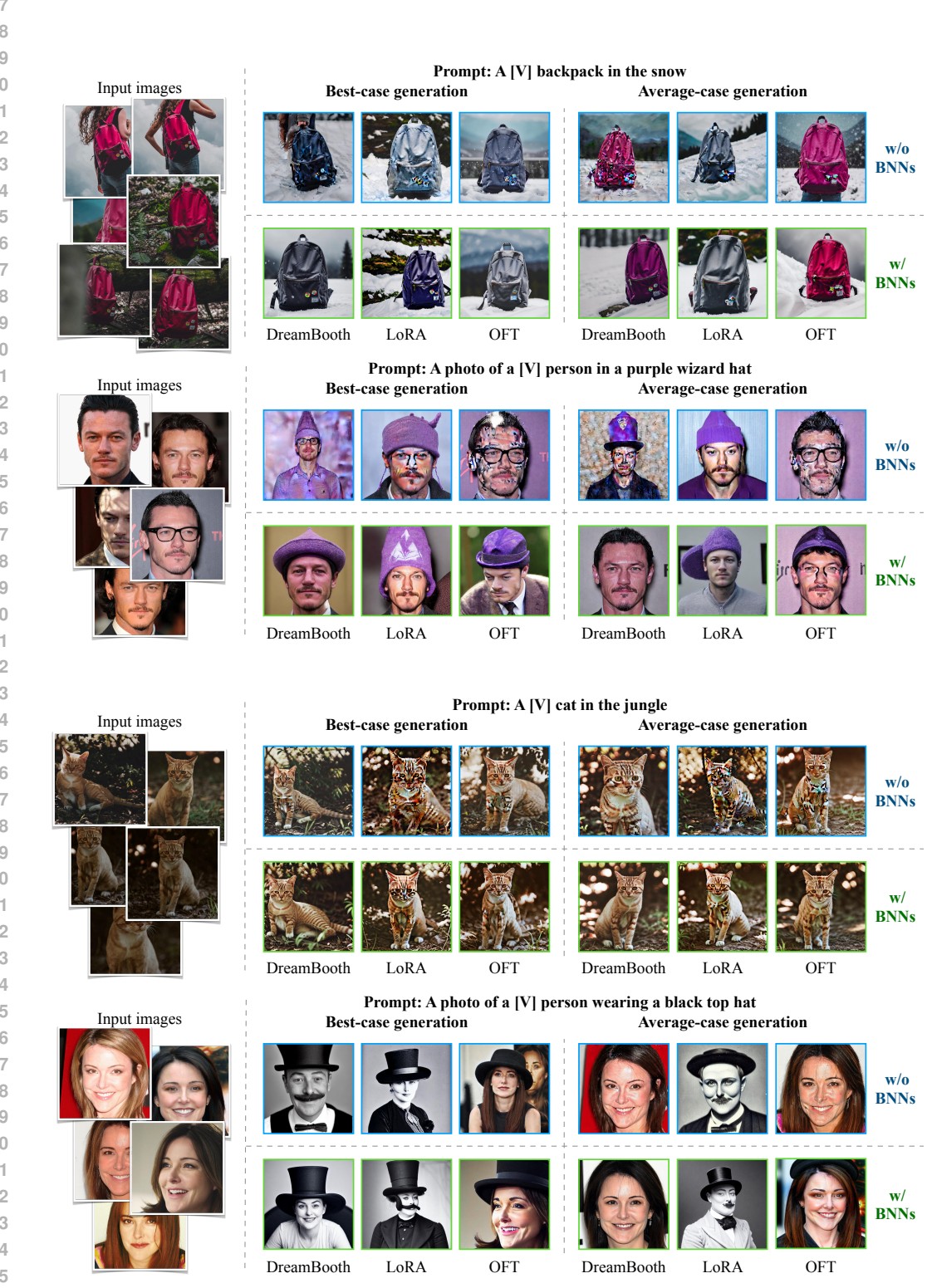

Figure 14: More visualizations on subject-driven and object-driven scenarios.

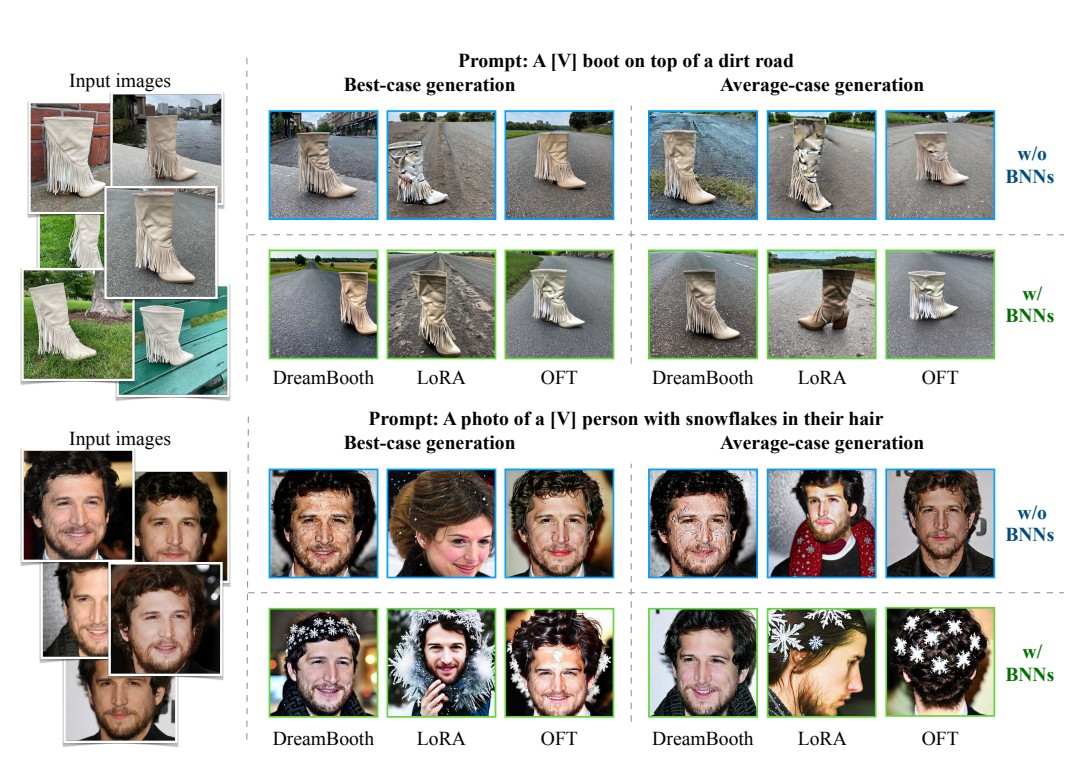

Figure 15: More visualizations on subject-driven and object-driven scenarios.

