# OpenReview forum: "Exploring Diffusion Models' Corruption Stage in Few-Shot Fine-tuning and Mitigating with Bayesian Neural Networks"
_ICLR.cc/2025/Conference — Submitted to ICLR 2025_

### Official Review · Reviewer_753t · 2024-10-29

**Soundness:** 3
**Presentation:** 3
**Contribution:** 2
**Rating:** 6
**Confidence:** 4

**Summary:**

This paper presents an empirical observation of diffusion models during fine-tuning in a few-shot setting, where the generated image quality undergoes a “corruption stage.” The authors analyze this corruption phenomenon through a series of experiments, beginning with single-sample fine-tuning, and provide an explanation based on noise correction terms. Building on this, they propose using Bayesian Neural Networks (BNNs) to mitigate corruption across different fine-tuning mechanisms.

**Strengths:**

- The observed phenomenon is compelling and would be of significant interest to the community, especially as many works focus on fine-tuning pre-trained diffusion models for downstream tasks.
- The analytical experiments are well-designed and supported by reasonable explanations.
- The proposed use of BNNs consistently improves performance across various fine-tuning methods.

**Weaknesses:**

- This paper could benefit from a broader coverage and discussion of related works on the similar topic of generation dynamics, such as [a] and [b]. There are quite a few existing works that reveal relevant phenomena, despite that they tackle more broadly instead of focusing purely on fine-tuning scenarios.
- Fig. 4 is somewhat confusing; the text indicates that it shows the average value of $\sigma_{1,t}$ across different values of t, yet later mentions t=100. It’s unclear how t-values are defined in Fig. 4 and in the experiments, and how t specifically impacts the BNN mitigation effects
- The authors include qualitative examples of the best and average-case generations; however, I am curious about the worst-case scenarios. A detailed discussion of failure cases would be equally important and valuable, in many generation related works.
- Something more minor, I think it would be beneficial to compare the actual fine-tuning cost versus the BNNs applied tuning cost in Tab. 3 (or separately in Appendices).
- There is one missing "**(**" in Eq. 3. Also isn't it $KL(Q_w(\theta)||P(\theta))$ rather than $KL(P(\theta)||Q_w(\theta))$, or did I miss anything?

---
[a] Spontaneous symmetry breaking in generative diffusion models, NeurIPS 2023

[b] Dynamical Regimes of Diffusion Models, 2024.

**Questions:**

Please see the weaknesses for my detailed questions.

---

> ### Author Response · Authors · 2024-11-21
> **Official Comments by Authors**
>
> Thank you for your constructive comments and suggestions, which have been immensely helpful in enhancing our paper. Below, we first restate your comments, followed by our detailed responses to each point.
>
> > **Weakness 1:** *This paper could benefit from a broader coverage and discussion of related works on the similar topic of generation dynamics, such as [a] and [b]. There are quite a few existing works that reveal relevant phenomena, despite that they tackle more broadly instead of focusing purely on fine-tuning scenarios.*
>
> **Answer:** Thank you very much for pointing out these two papers. These papers discuss the generation dynamics of general diffusion models. As we observe the corruption stages only in the few-shot fine-tuning scenarios, we do not find a direct connection between these two papers and ours. We will continue to pay attention on this related field and leave the relevant discussion and potential improvement as interesting future work.
>
> > **Weakness 2:** *Fig. 4 is somewhat confusing; the text indicates that it shows the average value of $\sigma_{1,t}$ across different values of t, yet later mentions t=100. It’s unclear how t-values are defined in Fig. 4 and in the experiments, and how t specifically impacts the BNN mitigation effects*
>
>
> **Answer:** In our modeling, $\sigma_1$ is a constant intrinsic to the model itself. To estimate its value approximately, we employ an averaging method to minimize potential estimation inaccuracies and present the plot curve clearly.
>
>
> Later in the text, $t=100$ is introduced as a reference point for adding additional noise to demonstrate the emergence of the corruption stage. By applying the estimated $\sigma_1$ value to Eq. (2), we calculate the increasing scale of the corruption noise. Fig. 5 then presents experimental results showing an increasing trend that aligns with our calculations. These findings suggest that our formulation is consistent with real-world scenarios.
>
>
> >**Weakness 3:** *The authors include qualitative examples of the best and average-case generations; however, I am curious about the worst-case scenarios. A detailed discussion of failure cases would be equally important and valuable, in many generation related works.*
>
> **Answer:**
> 1. In existing studies, visual results either showcase random images (Qiu et al., 2023) or only the best outcomes (Ruiz et al., 2023), without specific selection criteria explained. We have selected our showcased examples based on defined quantitative metrics, presenting both average and best cases to ensure a thorough comparative analysis.
>
> 2. In practical scenarios, users typically generate multiple images and manually choose the most suitable one for use. Therefore, comparing the best cases essentially evaluates the quality of the chosen images, while the comparison of average cases assesses the difficulty of selecting an adequate image. Conversely, worst-case scenarios may hold less practical relevance, as users may regenerate images until achieving a satisfactory result, effectively bypassing such cases.
>
>
>
> >**Weakness 4:** *Something more minor, I think it would be beneficial to compare the actual fine-tuning cost versus the BNNs applied tuning cost in Tab. 3 (or separately in Appendices).*
>
> **Answer:** As shown in Tab. 3, we report the memory and time costs during fine-tuning, with the additional costs from applied BNNs remaining low. Furthermore, during inference, our method introduces no additional memory or time costs by using the mean value of BNN layers, as indicated in Lines 345-347.
>
>
> >**Weakness 5:** *There is one missing "(" in Eq. 3. Also isn't it $KL(Q_w(\theta)||P(\theta))$ rather than $KL(P(\theta)||Q_w(\theta))$?*
>
> **Answer:** We could not find the mentioned missing "(" in Eq. (3) and would appreciate further details. Thank you very much for pointing out the typo in $KL(P(\theta)||Q_w(\theta))$! We have corrected it in the uploaded version.

---

> > ### Comment · Reviewer_753t · 2024-11-24
> > **Thank you for the author responses**
> >
> > I have reviewed the author responses as well as the comments from other reviewers.
> >
> > Regarding my own comments, I believe it remains important and valuable to discuss failure and worst cases, even if they are not commonly addressed in many applied vision papers. Simply stating that "our baseline methods don't report them" does not justify omitting failure analysis in this particular work. Many applied vision papers do not sufficiently address failure cases, but this is a limitation we should aim to improve upon rather than follow, especially considering the tendency of many generative works to focus on "cherry-picked" results.
> >
> > I also find some comments from R-xPGM and R-c7ut quite insightful, particularly those addressing the theoretical justification of the corruption stage and the question of diversity.  I would like to adjust my final rating also based on their further feedback and discussion.

---

> ### Author Response · Authors · 2024-11-25
> **Updated result about worst-case generation**
>
> Thank you for pointing out the importance of avoiding cherry-picked results; we agree. To ensure a comprehensive evaluation, our quantitative comparisons have reported average performance across 100 images per class, covering best, average, and worst cases, while qualitative comparisons highlight representative best and average cases selected by clear criteria (Appendix Sec. G), as provided in Figs. 6 and 13–14.
>
> Nonetheless, building on the reviewer’s suggestion, we recognize that including worst-case examples enhances the thoroughness of the comparison and provides better insights related to our method and claims. In response, we present worst-case generations for DreamBooth fine-tuning with and without BNNs, identified using CLIP-IQA, as detailed in revised Appendix Sec. G and Fig. 12. The results show that DMs fine-tuned without BNNs usually present the corruption while BNNs successfully mitigate it, further supporting the effectiveness of applying BNNs during few-shot fine-tuning.

---

### Official Review · Reviewer_JZKK · 2024-11-01

**Soundness:** 2
**Presentation:** 2
**Contribution:** 2
**Rating:** 3
**Confidence:** 5

**Summary:**

The paper investigates the “corruption stage” in diffusion models during few-shot fine-tuning, where initial image quality improvements degrade as training progresses, leading to noise and overfitting. The authors propose using Bayesian Neural Networks (BNNs) to mitigate this by broadening the learned distribution, effectively improving the fidelity and diversity of generated images. BNNs add no additional inference cost and are compatible with current fine-tuning methods.

**Strengths:**

1. The use of BNNs to address noise and degradation in diffusion models’ fine-tuning is innovative, providing new insights and effective mitigation strategies for this issue.

2. The paper provides a robust theoretical foundation and demonstrates the effectiveness of BNNs through various empirical tests, showcasing the improvement in image quality and diversity.

3. BNNs integrate smoothly with existing fine-tuning frameworks, and by not increasing inference costs, they retain practical relevance for computationally constrained applications.

**Weaknesses:**

1. Although the study shows reasonable results within the specific task context, its testing is limited to the dataset used in DreamBooth, which is too small in scope. While the authors have experimented with fine-tuning using 1, 2, and 6 images, it remains unclear whether the observed phenomena ("corruption stage") would still occur when the dataset size is increased to 1000 images or when batch size is adjusted. Given approaches like ControlNet and IP-adapter, how can it be demonstrated that the issues observed by the authors persist in large-scale fine-tuning scenarios? Furthermore, while the authors express hope that their observations might inform future research on diffusion models, it is uncertain if the proposed use of BNNs remains effective in large-scale data fine-tuning settings.

2. The paper essentially identifies a new problem and applies BNNs to few-shot fine-tuning tasks such as LoRA, which makes the novelty of the contribution somewhat limited.

3. The experiments mainly focus on score-matching-based Stable Diffusion models. It is uncertain how the proposed approach would perform with the more recent flow-matching models or with models using DiT as the denoising backbone.

**Questions:**

See Weaknesses above.

---

> ### Author Response · Authors · 2024-11-21
> **Official Comments by Authors (1/2)**
>
> Thank you for your constructive comments and suggestions, which have been immensely helpful in enhancing our paper. Below, we first restate your comments, followed by our detailed responses to each point.
>
> > **Weakness 1-1:** *Although the study shows reasonable results within the specific task context, its testing is limited to the dataset used in DreamBooth, which is too small in scope.*
>
> **Answer:** DreamBooth is a classic dataset in the few-shot fine-tuning domain and is widely used in the literature [1,2]. Moreover, as shown in Sec. 5, we evaluate our approach not only on DreamBooth but also on CelebA-HQ, addressing both object-driven and subject-driven generation tasks. This diversity in tasks and datasets verifies the popularity of the corruption stage and demonstrates the generality of the proposed approach in few-shot fine-tuning.
>
>
> > **Weakness 1-2:** *While the authors have experimented with fine-tuning using 1, 2, and 6 images, it remains unclear whether the observed phenomena ("corruption stage") would still occur when the dataset size is increased to 1000 images or when batch size is adjusted. Given approaches like ControlNet and IP-adapter, how can it be demonstrated that the issues observed by the authors persist in large-scale fine-tuning scenarios? Furthermore, while the authors express hope that their observations might inform future research on diffusion models, it is uncertain if the proposed use of BNNs remains effective in large-scale data fine-tuning settings.*
>
> **Answer:** As discussed in Sec. 1, few-shot fine-tuning focuses on the task of personalizing generation based on a small set of training images. This approach is important and widely adopted, as evidenced by the tens of thousands of checkpoints available on platforms such as Civitai and HuggingFace, hence we focus on this kind of setting in this paper.
>
> In few-shot fine-tuning, the training set typically consists of fewer than 100 images, a scale specifically designed to be user-friendly for applying personalization, as supported by practical scenarios demonstrated on platforms like Civitai and HuggingFace. We adopt this setting, consistent with previous works such as DreamBooth [1] and OFT [2]. Consequently, large-scale data fine-tuning is beyond the scope of this paper. Instead, we focus on identifying the corruption stage and propose its mitigation as a compelling direction for future research.
>
> The selection of a batch size of 1 is a common practice in few-shot fine-tuning tasks, as noted by Ruiz et al. (2023) [1] and Qiu et al. (2023) [2], and we have adopted this configuration based on them. In our experience, variations in batch size do not significantly affect the outcomes.
>
> > **Weakness 2:** *The paper essentially identifies a new problem and applies BNNs to few-shot fine-tuning tasks such as LoRA, which makes the novelty of the contribution somewhat limited.*
>
> **Answer:** Few-shot fine-tuning has been an active area of research in the context of diffusion models since 2022, as outlined in Sec. 2 with a series of related works. Its importance is underscored by the ability for individuals to easily adapt large-scale models to their unique styles or specific photos of a given object. This approach has seen widespread use, with tens of thousands of checkpoints being released online.
>
> The reason why we focus on the potential challenges of few-shot fine-tuning is its extensive real-world applications and transformative impact. As this technique continues to shape personalization and style adaptation tasks, addressing its challenges becomes crucial for ensuring  its reliability in practical scenarios.
>
> [1] Nataniel Ruiz, Yuanzhen Li, Varun Jampani, Yael Pritch, Michael Rubinstein, and Kfir Aberman. DreamBooth: Fine Tuning Text-to-Image Diffusion Models for Subject-Driven Generation. In CVPR, 2023.
>
> [2] Zeju Qiu, Weiyang Liu, Haiwen Feng, Yuxuan Xue, Yao Feng, Zhen Liu, Dan Zhang, Adrian Weller, and Bernhard Schölkopf. Controlling Text-to-Image Diffusion by Orthogonal Finetuning. In NeurIPS, 2023.

---

> ### Author Response · Authors · 2024-11-21
> **Official Comments by Authors (2/2)**
>
> > **Weakness 3:** *The experiments mainly focus on score-matching-based Stable Diffusion models. It is uncertain how the proposed approach would perform with the more recent flow-matching models or with models using DiT as the denoising backbone.*
>
> **Answer:** Score-matching-based Stable Diffusion models are currently among the most prevalent methods for few-shot fine-tuning tasks in practice. Previous approaches in fine-tuning, such as Textual Inversion [1], DreamBooth [2], and OFT [3], have also concentrated on using these models. Consequently, we adopt their experimental settings in our study.
>
>
> To further verify the generality of our proposed approach, we present additional experiment results on a representative DiT-based diffusion model, i.e., SD v3, as the reviewer suggests. Due to the time and memory limitations, LoRA is applied as the baseline fine-tuning approach. The outcomes are detailed in the Table R3 below. Both fidelity as measured by Dino and image quality as assessed by Clip-IQA indicate decreasing trends, and the DM fine-tuned with BNNs demonstrates significant and consistent performance improvements. These results reinforce the correctness and generality of our analysis and proposed methodology.
>
>
> **Table R3(a): Comparison between SD v3 fine-tuned with and without BNNs on fidelity measured by Dino**
> | Steps        | 200   | 400   | 600   | 800   | 1000  | 1200  | 1400  | 1600 |
> | ------------ | ----- | ----- | ----- | ----- | ----- | ----- | ----- | ----- |
> | LoRA         | 0.521 | 0.485 | 0.490 | 0.465 | 0.424 | 0.398 | 0.398 | 0.373 |
> | LoRA w/ BNNs | 0.470 | 0.550 | 0.610 | 0.560 | 0.620 | 0.600 | 0.573 | 0.551 |
>
> **Table R3(b): Comparison between SD v3 fine-tuned with and without BNNs on image quality measured by Clip-IQA**
> | Steps        | 200   | 400   | 600   | 800   | 1000  | 1200  | 1400  |1600|
> | ------------ | ----- | ----- | ----- | ----- | ----- | ----- | ----- | ----- |
> | LoRA         | 0.843|0.860|0.816|0.779|0.749|0.803|0.617|0.637|
> | LoRA w/ BNNs | 0.840|0.839|0.838|0.816|0.792|0.784|0.771|0.731|
>
>
> [1] Rinon Gal, Yuval Alaluf, Yuval Atzmon, Or Patashnik, Amit H Bermano, Gal Chechik, and Daniel Cohen-Or. An Image Is Worth One Word: Personalizing Text-to-Image Generation Using Textual Inversion. arXiv preprint arXiv:2208.01618, 2022.
>
> [2] Nataniel Ruiz, Yuanzhen Li, Varun Jampani, Yael Pritch, Michael Rubinstein, and Kfir Aberman. DreamBooth: Fine Tuning Text-to-Image Diffusion Models for Subject-Driven Generation. In CVPR, 2023.
>
> [3] Zeju Qiu, Weiyang Liu, Haiwen Feng, Yuxuan Xue, Yao Feng, Zhen Liu, Dan Zhang, Adrian Weller, and Bernhard Schölkopf. Controlling Text-to-Image Diffusion by Orthogonal Finetuning. In NeurIPS, 2023.

---

> ### Author Response · Authors · 2024-11-29
>
> Dear reviewer JZKK,
>
> As there are only four days left in the discussion period, we hope you could please consider getting back to us. Was our response above satisfactory? If not, we would very much appreciate a chance to address further concerns before the end of the discussion period.
>
> Thank you again for your time!

---

### Official Review · Reviewer_c7ut · 2024-11-02

**Soundness:** 3
**Presentation:** 3
**Contribution:** 3
**Rating:** 6
**Confidence:** 3

**Summary:**

This paper investigates the phenomenon of the "corruption stage" observed in few-shot fine-tuning of Diffusion Models (DMs), where image fidelity initially improves but then deteriorates due to noisy patterns, before recovering with severe overfitting. Through heuristic modeling, the authors attribute this corruption stage to the narrowed learned distribution inherent in few-shot fine-tuning. To address this, they propose using Bayesian Neural Networks (BNNs) to expand the learned distribution via variational inference, thereby mitigating the corruption. Experimental results demonstrate that this method significantly enhances image fidelity, quality, and diversity in both object-driven and subject-driven generation tasks.

**Strengths:**

- This work identifies and thoroughly investigates the "corruption stage" in few-shot fine-tuning of DMs, and it presents an innovative solution using BNNs to expand the learned distribution.
- The rigorous experimental design compares different fine-tuning methods across object-driven and subject-driven tasks. Multiple quantitative metrics (e.g., Clip-T, Dino, Clip-I) validate the effectiveness of the proposed method, and user studies support these quantitative results.
- The paper follows a clear logical flow, beginning with an explanation of the corruption stage, followed by an introduction to the application and benefits of BNNs.

**Weaknesses:**

1. Most experiments focus on specific datasets like DreamBooth and the Stable Diffusion v1.5 model without testing other diffusion models or more advanced Stable Diffusion versions, such as SDXL, SD-v3/3.5. I understand that the authors may be trying to solve a fundamental problem, but whether this problem exists on more advanced models still needs to be proven.

2. While BNNs offer significant performance improvements, the paper lacks a detailed analysis of computational costs, such as memory usage and inference time, compared to traditional methods, which is essential for assessing applicability in real-world scenarios. In my experience, there is often a trade-off between computational cost and performance for BNN. Does this exist in this work? If it exists, could the author provide a quantitative comparison of memory usage and inference time between their BNN approach and traditional methods?

3. Although the paper claims that BNNs mitigate corruption and reduce overfitting, there is limited information on how BNNs dynamically balance image diversity and fidelity at different training stages. Could the authors provide plots or other metrics demonstrating this balance across various training stages?

**Questions:**

1. While BNNs reduce corruption, has the paper considered additional regularization methods to prevent overfitting in later stages?

2. Impact of Hyperparameter Tuning.

3. Does the use of BNNs significantly increase training time or memory consumption?

---

> ### Author Response · Authors · 2024-11-21
> **Official Comments by Authors (1/2)**
>
> Thank you for your constructive comments and suggestions, which have been immensely helpful in enhancing our paper. Below, we first restate your comments, followed by our detailed responses to each point.
>
> > **Weakness 1** *Most experiments focus on specific datasets like DreamBooth and the Stable Diffusion v1.5 model without testing other diffusion models or more advanced Stable Diffusion versions, such as SDXL, SD-v3/3.5. I understand that the authors may be trying to solve a fundamental problem, but whether this problem exists in more advanced models still needs to be proven.*
>
>
> **Answer:** Classical DMs like SD v1.5 are relatively popular and friendly in few-shot fine-tuning tasks given their less computation and memory costs. Therefore, we mainly focus on the experiments on SD v1.5 model, following previous work (Qiu et al., 2023; Ruiz et al., 2023). Nonetheless, we also evaluate our approach using both subject-driven and object-driven generation tasks across the DreamBooth and CelebA-HD datasets for comprehensive evaluation. We also test various diffusion models (DMs), specifically SD versions 1.4, 1.5, and 2.0. These results comprehensively assess the efficacy of our proposed method.
>
>
> To further explore the effectiveness of our approach on more advanced DMs as the reviewer suggests, additional experiments were conducted with SD v3, fine-tuned both with and without BNNs. Due to the time and memory limitations, LoRA is applied as the baseline fine-tuning approach. We indeed observed that the corruption stage is alleviated to some extent on this more advanced DM. However, BNNs are still useful under this scenario, as detailed in the Table R1 below. Both fidelity as measured by Dino and image quality as assessed by Clip-IQA indicate decreasing trends, and the DM fine-tuned with BNNs demonstrates significant and consistent performance improvements. These results clearly demonstrate the generality of our analysis and proposed methodology.
>
>
> **Table R1(a): Comparison between SD v3 fine-tuned with and without BNNs on fidelity measured by Dino**
> | Steps        | 200   | 400   | 600   | 800   | 1000  | 1200  | 1400  | 1600 |
> |------------ | ----- | ----- | ----- | ----- | ----- | ----- | ----- | ----- |
> | LoRA         | 0.521 | 0.485 | 0.490 | 0.465 | 0.424 | 0.398 | 0.398 | 0.373 |
> | LoRA w/ BNNs | 0.470 | 0.550 | 0.610 | 0.560 | 0.620 | 0.600 | 0.573 | 0.551 |
>
> **Table R1(b): Comparison between SD v3 fine-tuned with and without BNNs on image quality measured by Clip-IQA**
> | Steps        | 200   | 400   | 600   | 800   | 1000  | 1200  | 1400  |1600|
> | ------------ | ----- | ----- | ----- | ----- | ----- | ----- | ----- | ----- |
> | LoRA         | 0.843|0.860|0.816|0.779|0.749|0.803|0.617|0.637|
> | LoRA w/ BNNs | 0.840|0.839|0.838|0.816|0.792|0.784|0.771|0.731|
>
>
> > **Weakness 2 & Question 3:** *While BNNs offer significant performance improvements, the paper lacks a detailed analysis of computational costs, such as memory usage and inference time, compared to traditional methods, which is essential for assessing applicability in real-world scenarios. In my experience, there is often a trade-off between computational cost and performance for BNN. Does this exist in this work? If it exists, could the author provide a quantitative comparison of memory usage and inference time between their BNN approach and traditional methods?*
> >*Does the use of BNNs significantly increase training time or memory consumption?*
>
> **Answer:** For training, as shown in Tab. 3, the improvements achieved by applying BNNs are consistent, with relatively low additional memory and time costs.
>
> For inference, as indicated in Line 344 of Sec. 4.2, one can use the mean value of BNNs to generate images, resulting in no additional memory or time costs. These results confirm the effectiveness and efficiency of BNNs in few-shot fine-tuning tasks.

---

> ### Author Response · Authors · 2024-11-21
> **Official Comments by Authors (2/2)**
>
> > **Weakness 3:** *Although the paper claims that BNNs mitigate corruption and reduce overfitting, there is limited information on how BNNs dynamically balance image diversity and fidelity at different training stages. Could the authors provide plots or other metrics demonstrating this balance across various training stages?*
>
> **Answer:** Our primary focus in this paper is on identifying corruptions and mitigating them using BNNs. As such, detailed discussions on the dynamic management of image diversity and fidelity by BNNs through different training stages are not extensively covered. However, we have included an analysis of image fidelity trends during the fine-tuning process, which can be found in Figure 7(a). For image diversity, please refer to Table R2 below, which illustrates comparisons at different training iterations. These results demonstrate that models utilizing BNNs achieve improvement in both image diversity and fidelity, hence definitely reaching better balance between them.
>
>
> **Table R2: Diversity comparison measured by Lpips on the DreamBooth dataset with
> different training iterations per image.**
> | Training iterations per image    | 50    | 100   | 200   | 400   |
> | -------- | ----- | ----- | ----- | ----- |
> | w/o BNNs | 0.720 | 0.696 | 0.611 | 0.516 |
> | w/ BNNs  | 0.719 | 0.704 | 0.640 | 0.485 |
>
>
> > **Question 1:** *While BNNs reduce corruption, has the paper considered additional regularization methods to prevent overfitting in later stages?*
>
> **Answer:** Yes, our formulation inherently includes a regularization term. In our formulation of BNNs on DMs, the prior distribution is centered around the pre-trained parameters, as shown in Sec. 4.2. This configuration allows the regularization term, $L_r$, to effectively control the deviation between the fine-tuned and initial parameters, thus acting as a regularizer. Furthermore, as elaborated in Sec. 5.3, increasing the parameter $\lambda$ enhances generation diversity (measured by Lpips) but comes at the expense of image fidelity (evidenced by Dino metrics), which can be viewed as postponing the overfitting stage.
>
>
> > **Question 2:** *(The reviewer wonders the) Impact of Hyperparameter Tuning.*
>
> **Answer:** As discussed in Sec. 5.2-5.3, we experimented with various hyperparameter configurations including the number of training iterations, the number of training images, and specific parameters within BNNs, such as $\lambda$ the initialization value of $\sigma_\theta$. The outcomes demonstrate that BNNs effectively mitigate stages of corruption and enhance generation quality across a range of hyperparameter settings and exhibit the robustness to variations in internal hyperparameters.

---

> > ### Comment · Reviewer_c7ut · 2024-11-26
> >
> > Dear Authors,
> >
> > Thank you for the detailed responses. While I acknowledge the efforts in addressing my concerns and improving the manuscript, I feel the novelty and potential impact of the proposed method still align with my original score of 6.

---

> > > ### Author Response · Authors · 2024-11-27
> > > **Official Comments by Authors**
> > >
> > > Thank you for your reply and we are happy to see our response successfully addresses your concerns! We appreciate the time and efforts you have invested in reviewing our work. If any comments or questions arise in the future, please let us know.

---

### Official Review · Reviewer_xPGM · 2024-11-05

**Soundness:** 2
**Presentation:** 2
**Contribution:** 2
**Rating:** 5
**Confidence:** 4

**Summary:**

This paper aims to analyze the progress of few-shot fine-tuning of diffusion models and improve it. First, they present the empirical analysis on the few-shot fine-tuning, which shows the corruption stage with the abnormal decrease in the fidelity of generated images. Then, they propose heuristic modeling, that a marginal distribution of $x_0$ with a parameterized diffusion model $\theta$ is approximated by a multivariate Gaussian distribution, to explain the few-shot fine-tuning procedure. Based on this modeling, they tried to understand why the corruption stage occurs. Finally, they propose the Bayesian neural network (BNN) approach. Through their experiments, they show that their method improves various fine-tuning methods including Dreambooth, LoRA, and Orthogonal Finetuning.

**Strengths:**

- This work tries to analyze the foundational diffusion model’s fine-tuning process.
- The authors incorporate the user study to support the effectiveness.

**Weaknesses:**

**W1: Observed Issues with Incomplete Linkages**


1. **W1-1 Incomplete Rationale for “Finding a Sample to Minimize the Error Term”**
   - In line 229, the statement, *“pre-trained diffusion model finds a sample $( x^\star )$ to minimize the error term”*, appears potentially misleading. First, diffusion models do not involve an explicit search mechanism; the phrase *“finding a sample”* might imply a deliberate search process that does not actually occur within the diffusion framework.
   - Second, I question whether the experiments in Fig. 3 adequately support the claim that the diffusion model *“minimizes the error term.”* It seems the authors imply that an unchanged $ x_0 $ when $ x_t = 0 $ is an indicator of error minimization ($ \delta $). However, my understanding is that diffusion models generate $ x_0 $ as a memorized sample from the training dataset, rather than as a result of minimizing an error term. Specifically, two points should be considered:
      - **Corrupted State of $ x_t $:** If $ x_t $ is initialized in an out-of-distribution state, the pre-trained diffusion model may fail to denoise such samples effectively. This would challenge the claim that the model consistently generates a sample from the training data.
      - **Validation of Training Data:** It’s unclear whether the sample $ x_0 = 0 $ corresponds to a specific instance in the training data. The prompt used, *“a simple, solid gray image with no textures or variations,”* may merely cause the model to generate a gray image without textures, as prompted, rather than replicating a sample from the training dataset.


   **Suggestion:** The authors might consider initializing $ x_t $ with actual samples from the training dataset used to train the diffusion model, alongside corresponding text labels. If the model does not alter $ x_t $ in these cases and the pixel-wise differences between generated samples and training data remain small, this would better support the claim.


2. **W1-2 Lack of Theoretical Validation for Error Minimization in Fine-Tuning**
   - To establish that the fine-tuning process drives the error term $ \delta $ to zero, the authors should consider providing a theoretical analysis showing how diffusion losses are minimized. Without such a theoretical basis, it is challenging to understand why the heuristic modeling is effective or whether the error term $ \delta $ indeed approaches zero at the optimal point.


3. **W1-3 Incomplete Explanation for the Corruption Stage during Fine-Tuning**
   - Building on W1-2, without a theoretical analysis of how $ \delta $ changes throughout fine-tuning, it’s difficult to comprehend why the corruption stage occurs. According to the authors’ experiments, fine-tuning the diffusion model on a limited number of samples results in significant errors after several iterations. This raises questions about why naive fine-tuning leads to this corruption stage (i.e., why $ \delta $ is high and $ \sigma $ remains high), which remain unanswered by the current modeling and experimental outcomes. Consequently, I believe the cause of corruption is not thoroughly studied in this work, and the conclusion that *“after some fine-tuning stages, error exists due to limited training steps”* seems somewhat trivial.


4. **W1-4 Unclear Justification for Using Bayesian Neural Networks (BNNs)**
   - Extending from W1-3, without a clear understanding of the corruption stage, I am not fully convinced why a Bayesian Neural Network (BNN) is presented as the solution. The authors mention that *“the modeling of BNNs hinders the diffusion models from learning the exact distribution of the training dataset $ D $”* but do not elaborate sufficiently on why this is beneficial in mitigating the corruption stage. Additional clarification about the cause of the corruption stage and further explanation of how BNNs address this underlying issue are needed.

**Questions:**

Please see weakness

---

> ### Author Response · Authors · 2024-11-21
> **Official Comments by Authors (1/2)**
>
> Thank you for your constructive comments and suggestions, which have been immensely helpful in enhancing our paper. Below, we first restate your comments, followed by our detailed responses to each point.
>
> **W1-1 (Incomplete Rationale for "Finding a Sample to Minimize the Error Term")**
>
>
> **Answer:**
>
> 1. In Line 229, this is indeed a simplification to enable further discussion. The  experimental results in Lines 233–253 are used to demonstrate how this simplification aligns with real-world scenarios.
> 2. Regarding the corrupted state of $x_t$: If $x_t$ is out of distribution, such as having an excessively large magnitude, it can introduce too much noise for a diffusion model to fully denoise. However, in most inference cases, such as those using DDPM or DDIM, encountering such out-of-distribution $x_t$ is rare.
> 3. In our modeling and derivation in Eq. (2), the predicted original image $\hat{x}_0$ is proportional to the training image sample $x'$ when $x_t = 0$. This follows from the equation $\hat{x}_0 = x' + \delta_t = \left(1 - \frac{-\alpha_t \sigma_1^2}{\alpha_t \sigma_1^2 + (1 - \alpha_t)}\right)x'$ for any $t$. This property does not hold in other cases as long as $x_t \neq 0$ and $x_t \neq x'$. This is why we choose $x_t = 0$ for the experiment—it provides a case that can be visualized, as the model's output is in a clean form with a direct relation to the training sample $x'$. Experiments using data from the pretraining dataset, on the other hand, result in outputs that are not in a neat form, making them harder to visualize.
>
>
>
>
>
> **W1-2 (Lack of Theoretical Validation for Error Minimization in Fine-Tuning)**
>
>
>  **Answer:**
> In Appendix Sec. A, we have provided the proof for our modeling based on the assumption that the learned distribution is the same as the data distribution. According to the proof and Eq. (1), we can derive that the error term $\delta_t$ is zero at the optimal point where $\sigma_1 = 0$.
>
> Nonetheless, we provide a more direct proof for the error term approaches zero when the diffusion loss approaches to zero:
>
> **Proof.** As previous work points out (Ho et al., 2020), the probability of a diffusion model on a given distribution is lower bounded by its ELBO:
>
>
> $-\log P_{\theta}(x_{0}) \geq \mathbf{E_{q}}(x_{1:T} |x_{0}) \log \frac{ P_{\theta}(x_{0:T})}{Q(x_{1:T} |x_{0})},$
>
> and is simplified to diffusion loss: $L_{DM}(x_0):=\mathbf{E_{t, \epsilon\in N(0,1)}} \Vert\varepsilon_{\theta} (\sqrt{\alpha_{t}}x_0 + \sqrt{1-\alpha_{t}}\epsilon, t) -\epsilon\Vert^{2}$.
>
> The main target loss of fine-tuning methods is to directly minimize the diffusion loss for data within the fine-tuning data distribution. This can be interpreted as minimizing the KL divergence $\mathbf{E_{Q}}(x_{1:T}' | x_{0}') \log \frac{P_{\theta}(x_{0:T}')}{Q(x_{1:T}' | x_{0}')}$, where $Q(x_0')$ represents the fine-tuning data distribution.
>
>
>
> Under optimal conditions where $L_{DM}(x_0') \to 0$, we have $P_{\theta}(x_0') \to 1$ and $P_{\theta}(x_{t+1}' | x_{0}') \to Q(x_{t+1}' | x_{0}')$. This implies the identity of the learned $P$ and fine-tuned data distribution $Q$, hence we have
>
>
>    $$\arg\max P_\theta(x_0|x_t) = \arg\max Q(x_0|x_t) = x'.$$
>
> It corresponds to $\sigma_{1}=0$ in Eq. (1), leading to $\delta_{t}=0$.        □
>
> It verifies the correctness of our modeling from another side, hence further supports our theoretical analysis.
>
>
> **W1-3 (Incomplete Explanation for the Corruption Stage during Fine-Tuning)**
>
>  **Answer:** From Eq. (1), a high $\sigma_1$ leads to a larger error $\delta_{t}$, provided that the fine-tuned diffusion model's behavior adheres to our formulation. Our experiment in Fig. 3 demonstrates how the diffusion model aligns with our modeling under moderate training iterations. Furthermore, the results in Fig. 5 provide an example where the visualization aligns with the error calculated in Eq. (1). These experimental results all verify the rationality and correctness of our theoretical modeling and corresponding explanation.

---

> ### Author Response · Authors · 2024-11-21
> **Official Comments by Authors (2/2)**
>
> **W1-4 (Unclear Justification for Using Bayesian Neural Networks)**:
>
> **Answer:** In Sec. 4.1, we discuss the motivation for applying BNNs from two key perspectives: 1) The structure of BNNs hinders DMs from precisely capturing the training dataset's distribution, and 2) The inherent sampling randomness acts as an implicit form of data augmentation. Given the complexity of the mechanism of DNNs, the theoretical modeling of these principles is challenging. Nonetheless, we delve deeper into these two aspects and provide further discussions below.
>
> **Hindering Exact Distribution.** Without BNNs, the model outputs images with high probability, focusing on high-confidence cases. However, DMs trained with BNNs inherently generate images with lower probabilities, requiring the model to handle lower-confidence cases, effectively hindering the DMs from learning the exact distribution.
>
> **Implicit Data Augmentation.** Applying BNNs during training can be considered a form of implicit data augmentation. For instance, in a decoder-encoder structure (which is a common view of DNNs, especially in the U-Net architecture of DMs), applying BNNs to the encoder introduces perturbations in the encoding space. This acts as an augmentation at the encoding-space level, ultimately influencing the final decoded images without reducing the image quality, enhancing robustness and generalization.
>
> Moreover, our experiments significantly verify that using BNNs mitigates the corruption stages. As shown in Fig. 1, DMs fine-tuned with BNNs do not have the corruption stages while DMs fine-tuned without BNNs have obvious corruption stages. Quantitative evaluation results also verify the effectiveness of using BNNs to improve few-shot fine-tuning as shown in Sec. 5.

---

> > ### Comment · Reviewer_xPGM · 2024-11-25
> >
> > Thank you for providing detailed responses to my concerns. While several issues have been clarified, a few points still leave me hesitant to recommend acceptance. Below are my remaining concerns and suggestions for improvement:
> >
> >
> > - **W1-1**
> >    - I appreciate the explanation of why the authors chose $x_t=0$ for their experiments, as it provides a clean case for visualization and aligns with the theoretical framework. However, this choice seems to represent a highly specific scenario that may not be generalizable well to a broader case. Therefore, in this case, the practicality of their proposed modeling and theory can not be regarded as sufficient. I would like to listen to the author's thoughts regarding my claim.
> >
> > - **W1-2**
> >     - I really appreciate the author's response regarding the theory. I suggest to include the derivation for this comment.
> >
> > - **W1-3**
> >     -  Apologies for any confusion earlier. I would like to see an analysis of how the error term $\delta_t$ evolves during the iteration as the diffusion loss is minimized. While empirical studies can show the changing trends of $\delta_t$, this appears somewhat trivial, as iterative optimization naturally ensures reducing trends. What I am more interested in is how $\delta_t$ changes in theoretical setups. This would provide deeper insights into the relationship between $\delta$ and the optimization process and reinforce the theoretical justification for your model.

---

> > > ### Author Response · Authors · 2024-11-28
> > > **Further Comments by Authors (2/2)**
> > >
> > > **Reviewer:** *Apologies for any confusion earlier. I would like to see an analysis of how the error term $\delta_t$ evolves during the iteration as the diffusion loss is minimized. While empirical studies can show the changing trends of $\delta_t$, this appears somewhat trivial, as iterative optimization naturally ensures reducing trends. What I am more interested in is how $\delta_t$ changes in theoretical setups. This would provide deeper insights into the relationship between $\delta$ and the optimization process and reinforce the theoretical justification for your model.*
> > >
> > > **Answer:**
> > > Thank you for your insightful question regarding the evolution of $\delta_t$ during optimization. We would like to first clarify that our empirical studies (Fig. 4) show the reducing trends of $\sigma_1$ instead of $\delta_t$, and the iterative optimization only ensures reducing trends of $\sigma_1$ while cannot explain the abnormal increasing of $\delta_t$ during fine-tuning (i.e., the corruption stage). In fact, $\delta_t$ estimates the difference between the predicted image and the "expected" output image, which directly corresponds to the observed corruption stage. As the emergence of corruption stage is surprising and non-trivial, we are the first to study and understand this phenomenon and propose a well-designed modelling to successfully guide the application of BNNs to mitigate it and improve generation.
> > >
> > > Concretely, as shown in Eq. (1), $\delta_t$ is composed of two factors: one is related to $\sigma_1$, which decreases during fine-tuning, and the other is $(x_t - \sqrt{\alpha_t} x')$, related to the learned image set $\mathcal{I}_\theta$. Because of the inherent complexity of DMs and DNNs, it is difficult to accurately characterize how these two factors evolve during the whole fine-tuning process in theory. Therefore, we use empirical results and heuristic modeling to estimate these two factors and provide an explanation of the abnormal increasing $\delta_t$ during fine-tuning (i.e., the corruption stage), as shown in Sec. 3.3. Our explanation aligns well with experimental observations and effectively guides the application of BNNs to mitigate corruption. We hope our findings and proposed modeling approach could inform further research in this domain.

---

> > > > ### Comment · Reviewer_xPGM · 2024-11-28
> > > >
> > > > Thank you for your responses. However, after reviewing your replies, I feel that my original concerns have not been fully addressed. In particular:
> > > >
> > > > 1. The generalizability of choosing $x_t = 0$ still seems insufficiently justified. While your explanation highlights its simplicity, my concern about its practical applicability to broader cases remains unresolved.
> > > >
> > > > 2. The analysis of $\delta_t$ evolution remains largely empirical, without adequately addressing my concerns for a theoretical explanation or derivation.
> > > >
> > > > I believe these points are critical for supporting your proposed method. I hope you can provide further clarification or justification.

---

> > > > > ### Author Response · Authors · 2024-11-29
> > > > >
> > > > > Thank you for your prompt response and for highlighting these concerns. We would like to provide further clarification and justification as follows:
> > > > >
> > > > > 1. **On the generalizability of choosing $x_t = 0$:**
> > > > >    The use of $x_t = 0$ in our experiments was not intended as a universal theoretical framework but rather as experimental evidence to support our modeling in this manuscript. Therefore, concerns about its generalizability are not directly applicable. We acknowledge that $x_t = 0$ is unlikely to occur in practical applications. However, it just serves as a simplified case to validate our modeling and confirm the rationality of our analysis.
> > > > >
> > > > >    To further strengthen our validation, we have supplemented our results with experiments where $x_t = kx'$ for different $k$s (scaled versions of $x'$) as outlined in our previous reply. The results from these additional experiments also align well with our theoretical modeling, further supporting the rationality of our proposed framework.
> > > > >
> > > > > 2. **On the theoretical explanation of $\delta_t$ evolution:**
> > > > >    We understand the desire for a deeper theoretical analysis. To the best of our knowledge, current literature on diffusion models primarily discusses their properties at convergence, with limited exploration of the training dynamics. As such, a complete theoretical framework for analyzing these dynamics remains an open challenge.
> > > > >
> > > > >     Furthermore, our work is not aimed at providing a comprehensive theoretical derivation for these dynamics. Instead, the main contributions of our paper lie in identifying the problem of the corruption stage, proposing a simple yet rational explanation, and developing a practical solution. While we appreciate the value of deeper theoretical modeling, it is not the primary focus of this work.
> > > > >
> > > > >    Further exploration and modeling of the fine-tuning process represent valuable future directions. We hope our work can inspire more in-depth investigations in this area. If you have additional questions or theoretical suggestions, we welcome further discussion to enhance the understanding and impact of this work.

---

> ### Author Response · Authors · 2024-11-28
> **Further Comments by Authors (1/2)**
>
> Thank you for your further constructive comments! We greatly appreciate the time and effort you have dedicated to reviewing our work. Below, we respond to each point:
>
> **Reviewer:** *I appreciate the explanation of why the authors chose $x_t=0$ for their experiments, as it provides a clean case for visualization and aligns with the theoretical framework. However, this choice seems to represent a highly specific scenario that may not be generalizable well to a broader case. Therefore, in this case, the practicality of their proposed modeling and theory can not be regarded as sufficient. I would like to listen to the author's thoughts regarding my claim.*
>
> **Answer:** Thank you for your valuable feedback regarding our choice of setting $x_t = 0$. We would like to clarify the reasons behind this selection for verification and present broader new validation experiments.
>
>
> 1. According to our modeling, we could  **simply predict the corresponding prediction of $x_t = 0$ for both pretrained DMs and fine-tuned DMs**, hence providing direct support for our analysis and modeling. Concretely,
>     - In fine-tuned DMs, when setting $x_t = 0$, we have the predicted image $\hat{x_0} = (1-\frac{\sqrt{\alpha_t} \sigma_1^2}{\alpha_t \sigma_1^2+\left(1-\alpha_t\right)}\sqrt{\alpha_t}) x^{\prime}$. As $\alpha_t$ and $\sigma_1$ are both scalars, $\hat{x_0}$ is just a scaling of $x'$. Therefore, in visualization, we could observe an image similar to the training image $x'$.
>     - On the other hand, for a typical DM (i.e., a pretrained DM), $x_t = 0$ is within its learned distribution $\mathcal{I}_\theta$. Therefore, the corresponding $x'$ here is exactly $x' = x_t = 0$, hence it should predict $\hat{x_0} = x_t = x' = 0$ according to Eq. (2).
>
>     These discussions are successfully verified by experiments shown in Fig. 3, hence supporting our analysis and modeling.
>
> 2. In contrast, **using another general $x_t$ would introduce additional complexity to predict the output**, hence is difficult to be applied to support our analysis. For example, for a random non-zero $x_t$,
>
>     - The predicted image $\hat{x_0}$ of a fine-tuned DM would be a linear combination of the training image $x'$ and the random $x_t$ according to Eq. (2): $\hat{x_0} = (1-\frac{\sqrt{\alpha_t} \sigma_1^2}{\alpha_t \sigma_1^2+\left(1-\alpha_t\right)}\sqrt{\alpha_t}) x^{\prime} + \frac{\sqrt{\alpha_t} \sigma_1^2}{\alpha_t \sigma_1^2+\left(1-\alpha_t\right)}x_t$. Therefore, the generated image would be relatively complex and contain certain noises, hence cannot serve as a good visualization to support our analysis and modeling.
>     - The predicted image $\hat{x_0}$ of a pretrained DM depends on which sample $x^* \in \mathcal{I}_\theta$ minimizes the error term $\delta_t\left(x_t, x^*\right)$. As $x_t$ is random, we cannot know the concrete value of $x^*$ beforehand, making it difficult to predict the generated image.
>
>     In conclusion, using another random $x_t$ would make both visualization and comparison difficult. Therefore, we do not use them in our paper.
>
> 3. To provide further verification on our modeling, we present **more results with $x_t = kx'$** for different $k$s in revised Appendix Sec. F. According to our modeling in Eq. (2), a fine-tuned DM should predict the original image $\hat{x_0} = (1 + \frac{\sqrt{\alpha_t} \sigma_1^2}{\alpha_t \sigma_1^2 + (1 - \alpha_t)} (k - \sqrt{\alpha_t}))x'$, which is a scaling of $x'$. Experimental results shown in Fig. 12 support our analysis, hence further confirming the rationality of our modeling.
>
>
> We hope that these explanations help clarify the rationale behind the design. We would be more than happy to discuss further if necessary.
>
> **Reviewer:** *I really appreciate the author's response regarding the theory. I suggest to include the derivation for this comment.*
>
> **Answer:**  Thank you very much for your acknowledgement on our theory. Following your suggestion, we have included this proof in the revised Appendix Sec. B.

---

### Meta-Review · Area_Chair_utpQ · 2024-12-20

**Metareview:**

This paper addresses the corruption stage in few-shot fine-tuning of diffusion models, where image quality degrades due to overfitting. It introduces a Bayesian Neural Network approach to improve the fidelity and diversity of generated images. The method is compatible with existing fine-tuning techniques like Dreambooth and LoRA and adds no extra inference cost.

Reviewers concerned the choice of $x_t=0$. The authors state that this assumption was made to "closely approximate practical scenarios." However, they fail to provide a convincing explanation of whether this assumption is valid. Additionally, questions remain regarding the theoretical justification of the corruption stage and the matter of diversity.

**Additional Comments On Reviewer Discussion:**

The authors engaged in in-depth discussions with the reviewers, incorporating new experimental results, such as best and worst cases, to provide a more comprehensive presentation. I agree with the authors’ perspective on the algorithm settings and the concerns about large-scale experiments. However, some important technical details and aspects of correctness remain unclear.

---

### Decision · Program_Chairs · 2025-01-22

Reject